# TaCo: A Benchmark for Lossless and Lossy Codecs of Heterogeneous Tactile Data

**Zhengxue Cheng**[1]* **Yan Zhao**[1]  **Keyu Wang**[1]  **Hengdi Zhang**[2]  **Li Song**[1]

[1] Shanghai Jiao Tong University, Shanghai, China
[2] Paxini Tech., Shenzhen, China
zxcheng@sjtu.edu.cn

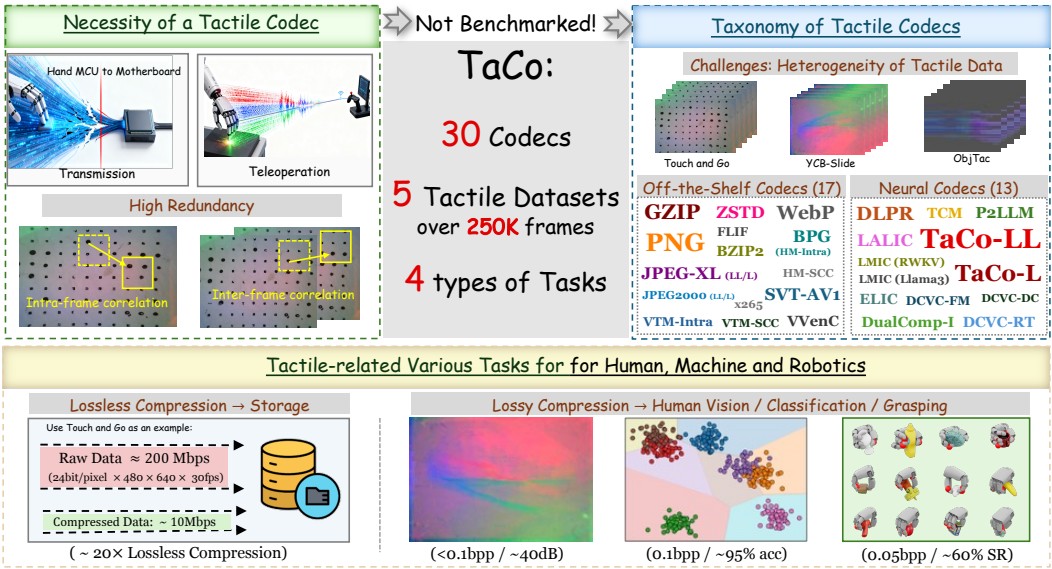

Figure 1: The motivation of our **TaCo** benchmark, established through an extensive evaluation on tactile codecs across multiple dimensions. First, we assess 30 off-the-shelf and neural codecs on 5 heterogeneous tactile datasets with more than 250K frames. Second, we introduce purely-trained TaCo-LL and TaCo-L codecs to explore the data-driven approaches in the field of lossless and lossy tactile data compression. Finally, we evaluate the coding performance on 4 distinct task types designed to serve for human, machine, and robotics.

## Abstract

Tactile sensing is crucial for embodied intelligence, providing fine-grained perception and control in complex environments. However, efficient tactile data compression, which is essential for real-time robotic applications under strict bandwidth constraints, remains underexplored. The inherent heterogeneity and spatiotemporal complexity of tactile data further complicate this challenge. To bridge this gap, we introduce **TaCo**, the first comprehensive benchmark for **Ta**ctile data **Co**decs. TaCo evaluates 30 compression methods, including off-the-shelf compression algorithms and neural codecs, across five diverse datasets from various sensor types. We systematically assess both lossless and lossy compression schemes on four key tasks: lossless storage, human visualization, material and object classification, and dexterous robotic grasping. Notably, we pioneer the development of data-driven codecs explicitly trained on tactile data, **TaCo-LL** (lossless) and **TaCo-L** (lossy). Results have validated the superior performance of our TaCo-LL and TaCo-L. This benchmark provides a foundational framework for understanding the critical trade-offs between compression efficiency and task performance, paving the way for future advances in tactile perception.

---

*Corresponding Author

# 1 INTRODUCTION

The acquisition and interpretation of tactile data are paramount for advancing embodied AI and achieving sophisticated human-machine interaction, as they provide the rich, physical context necessary for dexterous manipulation and awareness of the environment. However, the high-dimensional, spatio-temporally dense nature of tactile sensing results in rapidly growing data volumes, posing a significant bottleneck for real-time applications. Consequently, efficient tactile data compression is critical for real-time haptic feedback in dexterous hands, remote teleoperation, and large-scale storage of physical interactions for robotic model training.

While the imperative for efficient tactile data compression is well-established, current approaches remain diverse and fragmented. A corpus of existing research has explored this challenge through classical signal processing techniques (like dimensionality reduction and wavelet transforms), leveraging transforms and codecs designed for speech or image data. In recent years, data-driven methods have gradually gained popularity for their ability to learn optimal compact representations. Specifically, neural networks can learn compact latent representations in a data-driven manner, enabling efficient lossless or lossy compression (Liu et al., 2023b; Mentzer et al., 2019). Compared to traditional codecs, neural compression offers greater flexibility and adaptability, particularly in scenarios involving complex or irregular signal structures (Ma et al., 2019). These methods have been successfully applied in domains such as video and image compression (Zhao et al., 2025; Feng et al., 2025a), but they are still unexplored for tactile data. Another difficulty in generating a data-driven codec for tactile datasets is the heterogeneity, arising from different sensing principles: visuo-tactile sensors such as Gelsight (Yuan et al., 2017a) and DIGIT (Lambeta et al., 2020) capture surface transformation, while other force-based sensors (Paxini, 2025) measure force data. Therefore, the establishment of a comprehensive and open benchmark, comprising representative datasets, standardized evaluation metrics, and baseline models, is not merely beneficial but a necessary prerequisite for catalyzing advancements in this critical domain and enabling new research.

As illustrated in Fig. 1, we construct a comprehensive benchmark to evaluate the compression performance of various codecs on heterogeneous tactile datasets. First, we collect five diverse tactile datasets, and 30 representative codecs. They include off-the-shelf codecs designed for text, image and video, aiming to remove the 1D and 2D, inter-frame and intra-frame redundancies. We also incorporate neural codecs pretrained on other modalities to assess their cross-domain generalization on tactile data. Second, we propose two data-driven codecs, i.e. TaCo-LL and TaCo-L, which are trained from tactile datasets to learn intrinsic data patterns and exploit the redundancies in heterogeneous tactile data. Third, we evaluate tactile compression performance using four types of tasks: lossless compression, lossy compression for human perception, semantic classification, and robot grasping. Experimental results validate the superior performance of our proposed data-driven models, TaCo-LL and TaCo-L, and we hope our benchmark will inspire further research in this field.

In summary, our main contributions are as follows.

- We propose **TaCo**, the first comprehensive benchmark for tactile data codecs. It comprises five publicly tactile datasets, 30 codecs, and four tactile-related tasks: lossless compression, lossy compression for human visualization, tactile classification, and robotic grasping.

- We introduce **TaCo-LL** and **TaCo-L**, the first purely data-driven tactile codecs, trained end-to-end on heterogeneous tactile datasets to learn the intrinsic data distributions.

- Extensive experimental results demonstrate that our proposed TaCo-LL and TaCo-L models surpass existing methods across all four tasks, establishing a foundation for future research in the field of tactile data compression.

# 2 RELATED WORK

**Tactile Datasets.** Tactile datasets play a key role in advancing robotic perception and manipulation, supporting tasks like grasping, object recognition, and material classification. Several recent datasets focus only on tactile signals (Liu & Ward-Cherrier, 2024; Zhao et al., 2024; Suresh et al., 2023; Yuan et al., 2018; Higuera et al., 2024; Schneider et al., 2025). TIP Bench (Liu & Ward-Cherrier, 2024) converts sensor outputs into heatmaps and evaluates spatial acuity, stability, and generalization. FoTa (Zhao et al., 2024) merges multiple open datasets into a unified collection of

over three million samples. YCB-Slide (Suresh et al., 2023) records both simulated and real sliding interactions between a DIGIT tactile sensor and YCB objects. TacBench (Higuera et al., 2024) comprises 180,000+ unlabeled tactile images from surface-sliding interactions, facilitating large-scale self-supervised learning. Tactile MNIST (Schneider et al., 2025) provides both simulated and real GelSight interactions for MNIST digits, including 13,580 3D-printable meshes and 153,600 tactile recordings. ActiveCloth (Yuan et al., 2018) comprises 6,616 robotic squeeze trials on 153 garments, recording synchronized GelSight tactile videos and Kinect depth with 11 attributes.

Beyond pure tactile sensing, some other recent works (Yang et al., 2022; Feng et al., 2025b; Liu et al.; Cheng et al., 2025a; Yu et al., 2024; Suresh et al., 2024; Kerr et al., 2022; Yuan et al., 2017b; Li et al., 2019; Gao et al., 2021; Cheng et al., 2025b) incorporate multi-modal signals, combining tactile with vision, language, or audio to support cross-modal learning. For instance, Touch and Go (Yang et al., 2022) is the first large-scale tactile dataset collected in outdoor environments, capturing human interactions with natural objects via synchronized tactile and video data. VTDexManip (Liu et al.) provides 565,000 frames of video–tactile data from human multi-finger manipulations across 10 tasks and 182 objects, filling a gap in dexterous interaction datasets. Touch100K (Cheng et al., 2025a) compiles and cleans TAG and VisGel data into 100,147 high-quality triplets, offering the first large-scale alignment across tactile, visual, and linguistic modalities. TacQuad (Feng et al., 2025b) integrates four visual–tactile sensors, recording aligned tactile signals, RGB frames, and GPT-generated textual descriptions for multimodal reasoning. ObjectForlder (Gao et al., 2021) provides 100 neural object representations that encode 3D shape, appearance, sound, and tactile properties, supporting on-demand multimodal data generation for unified perception and control.

**Tactile Compression.** While tactile sensing continues to advance rapidly in resolution, sampling rate, and coverage, compression research for tactile data remains under-explored. Existing work has proposed some sparse or task-specific compression approaches (Hollis et al., 2016; Bartolozzi et al., 2017; Hollis et al., 2018; Shao et al., 2020; Hassen et al., 2020; Seeling et al., 2021; Liu et al., 2023a; Slepyan et al., 2024; Li et al., 2025; Lu et al., 2025). For instance, Shao et al. (2020) exploits the propagation of mechanical waves during dynamic contact to enable compact tactile encoding. (Hassen et al., 2020) proposes a perceptual vibrotactile codec that combines sparse linear prediction with an acceleration sensitivity function. (Seeling et al., 2021) achieves real-time tactile compression by combining bit-level truncation with delta-coding driven by just-noticeable-difference thresholds. Others like (Liu et al., 2023a) and (Slepyan et al., 2024) investigate dimensionality reduction via stacked auto-encoders or wavelet sparsification. However, these methods typically focus on simple signal sparsity or quantization strategies, often lack rigorous compression metrics and are tailored to relatively narrow scenarios or limited generalizability. In fact, many common tactile signals can be naturally transformed into image-like formats, enabling the use of standard image or general-purpose compressors. This direction is appealing not only because these compressors are well-established and widely available, but also because they offer tunable configurations to trade off compression ratio and distortion, making them adaptable to diverse robotic applications. Yet, this perspective remains largely under-explored in the tactile domain. To fill this gap, this paper presents a comprehensive benchmark for tactile compression methods, aiming to provide practical guidance and spark future research into efficient tactile data compression.

## 3 TACTILE DATASETS AND COMPRESSION METHODS

### 3.1 TACTILE DATASETS

We benchmark tactile compression across five representative datasets: Touch and Go (Yang et al., 2022), ObjectFolder 1.0 (Gao et al., 2021), SSVTP (Kerr et al., 2022), YCB-Slide (Suresh et al., 2023), and ObjTac (Cheng et al., 2025b). These datasets span a range of sensor types (vision-based and force-based), resolutions (from $120 \times 160$ to $640 \times 480$), and data scales, as detailed in Table. 1. Depending on the sensor type, tactile data exhibit strong structural heterogeneity, along with complex spatiotemporal correlations and redundancy, as illustrated in Fig. 1.

Specifically, the GelSight-based datasets (Touch and Go and ObjectFolder) and DIGIT-based datasets (SSVTP and YCB-Slide) are collected using vision-based tactile sensors that operate by illuminating a deformable elastomer surface with micro-LED arrays and capturing its surface deformation through an internal camera. This process converts tactile interactions into sequences of

Table 1: Introduction of the utilized tactile datasets.

| Dataset | #Objects | #Frames | Resolution | Sensor |
|---|---|---|---|---|
| Touch and Go (Yang et al., 2022) | 3971 | 13.9K | $640 \times 480 \times 30Hz$ | GelSight (Yuan et al., 2017a) |
| ObjectFolder 1.0 (Gao et al., 2021) | 100 | 100K | $120 \times 160 \times 30Hz$ | GelSight (Yuan et al., 2017a) |
| SSVTP (Kerr et al., 2022) | 10 | 4.5K | $240 \times 320 \times 30Hz$ | DIGIT (Lambeta et al., 2020) |
| YCB-Slide Suresh et al. (2023) | 10 | 4.5K | $240 \times 320 \times 30Hz$ | DIGIT (Lambeta et al., 2020) |
| ObjTac (Cheng et al., 2025b) | 56 | 135K | $5 \times 12 \times 200Hz$ | Force Sensor (Paxini, 2025) |

RGB images or videos, enabling direct applications of image or video compression techniques. In contrast, the ObjTac dataset is collected using force-based tactile sensors. The sensor comprises $N = 60$ contact points across the contact surface, each measuring a 3D force vector. These measurements form a temporally structured sequence of force data. To enable efficient compression, we map each 3D force vector to an RGB pixel and temporally stack the force readings across a time duration $T$, generating images of resolution $T \times 60$.

## 3.2 TACTILE COMPRESSION METHODS

We establish a benchmark for two categories of tactile codecs: 1) *off-the-shelf codecs* based on conventional signal processing, originally designed for general-purpose or visual data, and 2) *neural codecs* that leverage neural networks to learn data patterns end-to-end. As tactile signals are natively or transformable into image or video formats (see Section 3.1), our evaluation of neural codecs includes both pretrained image codecs and, to our knowledge, the first data-driven codecs explicitly trained on tactile datasets.

### 3.2.1 OFF-THE-SHELF COMPRESSION METHODS

Typically, off-the-shelf compression methods have been historically designed for text, image and video data. These classical techniques are fundamentally rooted in signal processing principles, aiming to eliminate statistical, spatial, or temporal redundancies present in 1D or 2D data.

**General-Purpose Compression Methods.** We evaluate three general-purpose lossless compressors: gzip (Pasco., 1996), zstd (Meta., 2015), and bzip2 (Seward, 2000), which are designed to exploit 1D symbol redundancy using techniques such as dictionary coding (e.g., LZ77 in gzip and zstd), block-sorting transforms (e.g., Burrows-Wheeler in bzip2), and entropy coding.

**Image and Video Compression Methods.** When treating tactile data as images, we evaluate six standard image lossless codecs: PNG (Boutell, 1997), FLIF (Sneyers, 2015), WebP (Google, 2010), JPEG-XL (Team, 2021), JPEG2000 (ISO/IEC, 2000), and BPG (Bellard, 2014) (the intra-mode of HEVC/265 codec). These image-specific compressors remove 2D spatial redundancy in images via predictive coding, transform coding (e.g., DCT or wavelets), and context-based entropy coding.

In addition, we also evaluate six lossy image codecs: JPEG-XL (Team, 2021), JPEG2000 (ISO/IEC, 2000), as well as the intra-frame and screen content coding (SCC) modes of HM Sullivan et al. (2012) and VTM (Bross et al., 2021) (i.e., HM-Intra, HM-SCC, VTM-Intra, VTM-SCC).

To further address the inter-frame redundancy in tactile data, we evaluate three off-the-shell video codecs, VVenC (Bross et al., 2021), x265 (Sullivan et al., 2012), and SVT-AV1 (Han et al., 2021). Due to the huge amount of video data, lossless video compression is rarely used in practice, so we only discuss lossy video compression methods.

### 3.2.2 NEURAL COMPRESSION METHODS

Recently, neural codecs have surpassed conventional codecs on text, image and video, owing to powerful learning capabilities of neural networks to fit the latent data distribution. However, tactile data exhibit unique statistical patterns, and heterogeneous tactile datasets involve different distributions, potentially making pre-trained neural encoders less applicable. Herein, we briefly introduce the diagram of learning-based lossy and lossless neural codecs, as Fig. 2.

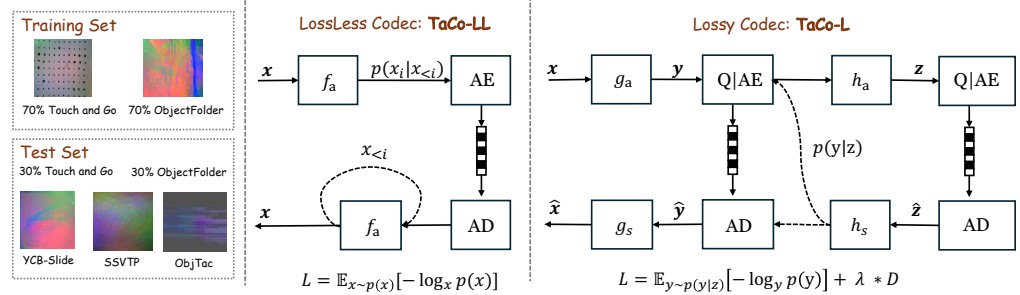

Figure 2: Diagram of data-driven compression methods and our proposed TaCo-L and TaCo-LL.

In the case of lossless compression, the tactile signal $x$ is sequentially fed into a neural network $f_a$ to predict the distribution of next symbol, $p(x_i|x_{<i})$, then it is followed by an arithmetic encoder (AE) to generate bitstream. The loss is the entropy, which is the minimal bound to encode $x$:

$$\mathcal{L} = \mathbb{E}[-\log_2(p(x_i|x_{<i}))] \tag{1}$$

At the decoder side, the symbols can be lossless autoregressively decoded through an arithmetic decoder (AD) and the same network.

For neural lossy codecs, the tactile signal $x$ is transformed through a transform function $g_a$ into a latent presentation $y$. Afterwards, $y$ is quantized through $Q$ to get discrete values $\hat{y}$, then it is followed by AE to generate bitstream. At the decoder side, $\hat{y}$ is decoded from bitstream using an AD and then transformed back to reconstructed images $\hat{x}$ though an inverse transform function $g_s$. $h_a$ and $h_s$ denote analysis and synthesis transforms in the hyper autoencoder to generate side bits $z$, as a prior to estimate density model of $\hat{y}$. The loss is defined as a rate-distortion function:

$$\mathcal{L} = \lambda \times \mathcal{D}(x, \hat{x}) + \mathbb{E}[-\log_2(p_{\hat{y}|\hat{z}}(\hat{y}|\hat{z}))] \tag{2}$$

where $\lambda$ is a hyper-parameter to control the bitrate, and all the parameters are learnable.

**Pretrained Neural Codecs**   The lossless neural-based methods include 5 image compression models, DLPR (Bai et al., 2024), P2LLM (Chen et al., 2024), and DualComp-I (Zhao et al., 2025), as well as LMIC (Deletang et al., 2024), a multi-modality compressor based on pretrained large language models (specifially, in this paper we use RWKV-7B (Peng et al., 2025) and Llama3-8B (AI@Meta, 2024) as LLMs for implementation). It is worth noting that these models are pretrained on natural language or image data, and evaluated on tactile data without any domain-specific adaptations.

For the purpose of lossy neural-based compression approaches, we evaluate a total of 6 compression models, consisting of three recent neural-based image codecs (ELIC (He et al., 2022), TCM (Liu et al., 2023b)) and LALIC (Feng et al., 2025a), and three recent neural-based video compressors (DCVC-DC (Li et al., 2023), DCVC-FM (Li et al., 2024), and DCVC-RT (Jia et al., 2025)).

**Tactile Data-Driven Neural Codecs**   To our knowledge, there have been yet no existing methods fully trained on tactile signals to explore the upper bound of tactile compression performance. To further explore the performance potential of data-driven codec, we retrain two state-of-the-art compression models, DualComp-I and LALIC, using tactile datasets. Specifically, DualComp-I operates lossless compression by tokenizing the input into discrete representations and applying an autoregressive model to predict each token's distributions, enabling efficient entropy coding. LALIC implements lossy compression and follows a variational auto-encoder (VAE) (Doersch, 2016) architecture. The two models are chosen for their competitive performance and efficiency in their respective domains. By retraining them on tactile data, we aim to assess the benefits of data-driven tactile compression. The retrained models are referred to as **TaCo-LL** (lossless) and **TaCo-L** (lossy), respectively, to distinguish them from their original pretrained versions.

Specifically, for **TaCo-LL**, the tokenization is conducted as shown in Fig. 3. We divide the input into $16 \times 16 \times 3$ patches to preserve local spatial correlations. We then flatten the data in a raster-scan order. For visuo-tactile data, including Touch and Go, YCB-slide, ObjectFolder, SSVTP, the RGB values are sequentially expanded as sub-pixels $(R_1, G_1, B_1, R_2, G_2, B_2, \cdots)$. For three-axis force

Figure 3: Detailed implementations of our proposed TaCo-L and TaCo-LL.

signals, i.e. ObjTac, are treated as three color channels and expanded as $(x_1, y_1, z_1, x_2, y_2, z_2, \cdots)$. For **TaCo-L**, we follow the setup of LALIC [1] and randomly crop or zero-pad the input tactile image to $256 \times 256$ resolution. Since the input tensor has three channels for both visuo-tactile data and force-tactile data, no tokenization is needed, as shown in Fig. 3. The network architecture is adopted from the LALIC model (Feng et al., 2025a) and the $g_a$ and $g_s$ consist of four downsampling and upsampling operations, respectively.

To this end, we benchmark a total of **30 codecs** to evaluate the compressibility of tactile data. Among them, 14 codecs (9 off-the-shell codecs, 4 neural codecs and one proposed TaCo-LL) support lossless compression, aiming to preserve exact signal fidelity. The remaining codecs (9 off-the-shell, 6 neural codecs and one proposed TaCo-L) are lossy, targeting higher compression ratios at the cost of some reconstruction distortion. These methods can also be categorized by their training data domain: 28 codecs are existing methods originally developed for general-purpose or visual data and applied without any tactile-specific adaptation, while the remaining two (TaCo-LL, TaCo-L) are data-driven models explicitly trained on tactile datasets. Evaluating the pretrained models allows us to assess how well existing compression techniques generalize to tactile data, whereas the tactile data-driven methods help explore the potential of tactile-aware compression strategies.

## 4 EXPERIMENTS

### 4.1 EXPERIMENTAL SETUP

We benchmark the performance of tactile compression on five representative tactile datasets: Touch and Go, ObjectFolder 1.0, SSVTP, YCB-Slide, and ObjTac, as shown in Section 3.1. Specifically, we randomly select 70% of the data from the Touch and Go and ObjectFolder datasets to train the TaCo-LL and TaCo-L models. The remaining 30% of the two datasets, together with the entire SSVTP, YCB-Slide and ObjTac datasets, are used for all methods' compression evaluation. Following (Zhao et al., 2025), for TaCo-LL we use the FusedAdam optimizer (NVIDIA, 2018) with a cosine annealing learning rate schedule Loshchilov & Hutter (2016), starting at $1 \times 10^{-4}$ and decaying to $2 \times 10^{-5}$ over 20 epochs. Following (Feng et al., 2025a), we train TaCo-L using the Adam optimizer (Kingma & Ba, 2014). The learning rate is set to $1 \times 10^{-4}$ for 40 epochs, then decayed to $1 \times 10^{-5}$ for another 4 epochs. The training is performed on two NVIDIA A100 GPUs.

### 4.2 LOSSLESS COMPRESSION

**Evaluation Metrics.** We evaluate lossless compression efficiency using bits per Byte, which quantifies the number of bits required to encode one byte of the original tactile data. Lower bits/Byte values indicate more effective compression, with uncompressed data corresponding to 8 bits/Byte. We also compare the complexity of different algorithms using four metrics, i.e. model parameters, MACs, inference speed (KB/s) on multiple devices (including NVIDIA A100 GPU, a MacBook Pro), and the frame per second (FPS) ranging with different spatial resolutions.

**Results.** Table. 2 benchmarks the lossless compression performance across five tactile datasets. As expected, all methods obviously reduce the data's storage cost, but the degree of compression varies across the compressors and datasets. Among off-the-shelf baselines, general-purpose compressors such as gzip and zstd achieve moderate compression ratios. Image-specific codecs like FLIF and JPEG-XL provide notably better results especially on vision-like tactile data, due to their ability to

---

[1] https://github.com/sjtu-medialab/RwkvCompress

Table 2: Comparison of lossless compression performance (bits/Byte) on five tactile datasets. The best results are highlighted in **bold blue**, second-best in **bold**, and third to fifth in underline. For TaCo, 12M/48M/96M denotes the model parameter. To show the compression performance more clearly, we also list the compression ratios relative to the uncompressed data (8 bits/Byte) in parentheses only for the best and second best results.

| | Compressor | bits/Byte↓ | | | | |
| | | TouchandGo | ObjectFolder | SSVTP | ObjTac | YCB-Slide |
|---|---|---|---|---|---|---|
| | uncompressed | 8 (1×) | 8 (1×) | 8 (1×) | 8 (1×) | 8 (1×) |
| Off-the-Shelf | gzip (Pasco., 1996) | 2.298 | 3.969 | 2.234 | 0.571 | 2.185 |
| | zstd (Meta., 2015) | 2.263 | 3.966 | 2.233 | 0.568 | 2.184 |
| | bzip2 (Seward, 2000) | 2.288 | 4.031 | 2.255 | 0.594 | 2.205 |
| | FLIF (Sneyers, 2015) | 0.808 (10×) | 3.765 | 1.567 | **0.363** (22×) | 1.489 (5×) |
| | BPG (Bellard, 2014) | 1.293 | 3.726 | 2.000 | 0.513 | 1.922 |
| | WebP (Google, 2010) | 0.936 | 3.612 | 1.820 | 0.424 (19×) | 1.767 |
| | JPEG-XL (Team, 2021) | 0.739 (11×) | 3.657 | 1.478 (5×) | 0.382 (21×) | 1.431 (6×) |
| | JPEG2000 (ISO/IEC, 2000) | 1.552 | 3.989 | 1.997 | 1.399 | 1.916 |
| | PNG (Boutell, 1997) | 2.500 | 3.964 | 2.233 | 0.579 | 2.183 |
| Neural | DLPR (Bai et al., 2024) | 1.082 | 3.774 | 1.539 | 0.522 | 1.503 |
| | P2LLM (Chen et al., 2024) | 1.212 | 3.400 | 1.804 | 0.546 | 1.512 |
| | Llama3* (Deletang et al., 2024) | 2.055 | 3.465 | 1.975 | 0.834 | 1.905 |
| | RWKV* (Deletang et al., 2024) | 2.223 | 3.718 | 2.010 | 0.540 | 1.880 |
| | DualComp-I (Zhao et al., 2025) | 0.948 | 3.126 (3×) | 1.442 (6×) | 0.540 | 1.388 (6×) |
| | TaCo-LL-12M (ours) | 0.622 (13×) | 3.098 (3×) | 1.457 (6×) | 0.569 | 1.520 |
| | TaCo-LL-48M (ours) | **0.504** (16×) | **2.923** (3×) | **1.249** (6×) | 0.411 (20×) | **1.321** (6×) |
| | TaCo-LL-96M (ours) | **0.447** (18×) | **2.709** (3×) | **1.066** (8×) | **0.360** (22×) | **1.073** (8×) |

Table 3: The complexity of lossless compression algorithms on five tactile datasets. † and ‡ indicates speeds measured on MacBook Pro CPU and NVIDIA A100 GPU, respectively.

| | Compressor | #Params↓ | MACs↓ | Speed (KB/s)↑ | Speed (FPS)↑ | | | | |
| | | | | | TouchandGo | ObjectFolder | SSVTP | ObjTac | YCB-Slide |
|---|---|---|---|---|---|---|---|---|---|
| Off-the-Shelf | gzip (Pasco., 1996) | - | - | 14500[†] | 15.7[†] | 252[†] | 62.9[†] | 190[†] | 63.9[†] |
| | zstd (Meta., 2015) | - | - | 11000[†] | 11.9[†] | 191[†] | 47.7[†] | 144[†] | 47.4[†] |
| | bzip2 (Seward, 2000) | - | - | 3300[†] | 3.58[†] | 57.3[†] | 14.3[†] | 43.3[†] | 14.3[†] |
| | FLIF (Sneyers, 2015) | - | - | 652[†] | 0.71[†] | 11.3[†] | 2.84[†] | 8.56[†] | 2.84[†] |
| | BPG (Bellard, 2014) | - | - | 180[†] | 0.20[†] | 3.13[†] | 0.78[†] | 2.36[†] | 0.78[†] |
| | WebP (Google, 2010) | - | - | 330[†] | 0.36[†] | 5.73[†] | 1.43[†] | 4.33[†] | 1.43[†] |
| | JPEG-XL (Team, 2021) | - | - | 970[†] | 1.05[†] | 16.8[†] | 4.21[†] | 12.7[†] | 4.21[†] |
| | JPEG2000 (ISO/IEC, 2000) | - | - | 5000[†] | 5.43[†] | 86.8[†] | 21.7[†] | 65.7[†] | 21.7[†] |
| | PNG (Boutell, 1997) | - | - | 200[†] | 0.22[†] | 3.47[†] | 0.87[†] | 2.63[†] | 0.87[†] |
| Neural | DLPR (Bai et al., 2024) | 22.3M | - | 640[‡] | 0.69[‡] | 11.1[‡] | 2.78[‡] | 8.41[‡] | 2.78[‡] |
| | P2LLM (Chen et al., 2024) | 8B | - | 20[‡] | 0.02[‡] | 0.35[‡] | 0.09[‡] | 0.26[‡] | 0.09[‡] |
| | Llama3* (Deletang et al., 2024) | 8B | 7.8G | 20[‡] | 0.02[‡] | 0.35[‡] | 0.09[‡] | 0.26[‡] | 0.09[‡] |
| | RWKV* (Deletang et al., 2024) | 7B | 7.2G | 86[‡] | 0.09[‡] | 1.49[‡] | 0.37[‡] | 1.13[‡] | 0.37[‡] |
| | DualComp-I (Zhao et al., 2025) | 96M | 59.9M | 317[‡] | 0.34[‡] | 5.50[‡] | 1.38[‡] | 4.16[‡] | 1.38[‡] |
| | TaCo-LL-12M (ours) | 12M | 11.6M | 614[‡] | 0.67[‡] | 10.7[‡] | 2.66[‡] | 8.06[‡] | 2.66[‡] |
| | TaCo-LL-48M (ours) | 48M | 33.3M | 360[‡] | 0.39[‡] | 6.25[‡] | 1.56[‡] | 4.73[‡] | 1.56[‡] |
| | TaCo-LL-96M (ours) | 96M | 59.9M | 317[‡] | 0.34[‡] | 5.50[‡] | 1.38[‡] | 4.16[‡] | 1.38[‡] |

exploit spatial correlations. Learning-based methods pretrained on natural images, such as DLPR, P2LLM, and DualComp-I, effectively capture intra-frame correlations in tactile signals and generally provide pleasing results. However, their performance remains limited by domain mismatch, especially on non-visual or structurally different tactile datasets.

To further explore the potential of data-driven compression, we retrain state-of-the-art lossless image compressor, DualComp-I, on tactile datasets and obtain our TaCo-LL model. The largest variant, TaCo-LL-96M, achieves the best performance across all five datasets, reaching 0.447 bits/Byte on TouchandGo, 2.709 bits/Byte on ObjectFolder, 1.066 on SSVTP, 0.360 bits/Byte on ObjTac, and 1.073 on TCB-Slide, corresponding to $18\times$, $3\times$, $8\times$, $22\times$, and $8\times$ compression ratios, respectively.

Table 4: Evaluation of lossy compression performance on five tactile datasets leveraging intra-frame compressors. The best results are shown in **blue bold**, the second-best in **bold**, and the third-best in underline. For the reference, the bandwidth consumption of the anchor HEVC-intra is approximately 2Mbps at the quality of 40dB, which is calculated by 0.22 bit per pixel $\times 640 \times 480 \times 30\text{fps} \times 10^{-6}$ for Touch and Go dataset, as Fig. 4.

| | Compressor | BD-Rate (%)↓ | | | | |
|---|---|---|---|---|---|---|
| | | TouchandGo | ObjectFolder | SSVTP | YCB-Slide | ObjTac |
| **Off-the-Shelf** | HM-Intra (Sullivan et al., 2012) | 0% | 0% | 0% | 0% | 0% |
| | HM-SCC (Sullivan et al., 2012) | -10.4% | 2.0% | 6.9% | 7.2% | **-44.5%** |
| | VTM-Intra (Bross et al., 2021) | -21.7% | **-19.7%** | **-16.0%** | **-24.4%** | -22.0% |
| | VTM-SCC (Bross et al., 2021) | -23.7% | -18.0% | -13.7% | -19.1% | **-44.3%** |
| | JPEG-XL (Team, 2021) | 66.7% | 60.6% | 77.5% | 96.9% | 99.4% |
| | JPEG2000 (ISO/IEC, 2000) | 59.7% | 69.9% | 107.9% | 89.1% | 103.8% |
| **Neural** | ELIC (He et al., 2022) | -40.2% | 0.6% | -5.8% | -9.2% | 44.5% |
| | LALIC (Feng et al., 2025a) | **-51.6%** | 0.2% | 4.3% | -4.6% | 32.8% |
| | TCM (Liu et al., 2023b) | -39.9% | 23.7% | 42.9% | 30.5% | 97.2% |
| | TaCo-L (Ours) | **-61.8%** | **-24.3%** | **-19.2%** | **-27.4%** | -27.0% |

Table 5: The complexity of lossy compression algorithms on five tactile datasets leveraging intra-frame compressors. † and ‡ indicates speeds measured on MacBook Pro CPU and NVIDIA A100 GPU, respectively.

| | Compressor | #Params↓ | MACs↓ | Speed (KB/s)↑ | Speed (FPS)↑ | | | | |
|---|---|---|---|---|---|---|---|---|---|
| | | | | | TouchandGo | ObjectFolder | SSVTP | YCB-Slide | ObjTac |
| **Off-the-Shelf** | HM-Intra (Sullivan et al., 2012) | - | - | 11.1[†] | 0.12[†] | 1.97[†] | 0.50[†] | 1.49[†] | 0.50[†] |
| | HM-SCC (Sullivan et al., 2012) | - | - | 31.3[†] | 0.34[†] | 0.56[†] | 0.41[†] | 0.42[†] | 0.41[†] |
| | VTM-Intra (Bross et al., 2021) | - | - | 9.22[†] | 0.10[†] | 1.63[†] | 0.41[†] | 1.23[†] | 0.41[†] |
| | VTM-SCC (Bross et al., 2021) | - | - | 4.61[†] | 0.05[†] | 0.72[†] | 0.18[†] | 0.55[†] | 0.18[†] |
| | JPEG-XL (Team, 2021) | - | - | 2305[†] | 2.50[†] | 40.0[†] | 10.0[†] | 30.2[†] | 10.0[†] |
| | JPEG2000 (ISO/IEC, 2000) | - | - | 13200[†] | 14.3[†] | 228[†] | 57.1[†] | 172[†] | 57.1[†] |
| **Neural** | ELIC (He et al., 2022) | 33.3M | 0.9M | 4075[‡] | 4.42[‡] | 70.7[‡] | 17.7[‡] | 53.5[‡] | 17.7[‡] |
| | LALIC (Feng et al., 2025a) | 63.2M | 0.7M | 3700[‡] | 4.01[‡] | 64.2[‡] | 16.1[‡] | 48.6[‡] | 16.1[‡] |
| | TCM (Liu et al., 2023b) | 75.9M | 1.8M | 5680[‡] | 6.16[‡] | 98.6[‡] | 24.7[‡] | 74.6[‡] | 24.7[‡] |
| | TaCo-L (Ours) | 63.2M | 0.73M | 3700[‡] | 4.01[‡] | 64.2[‡] | 16.1[‡] | 48.6[‡] | 16.1[‡] |

Table. 3 benchmarks the complexity of different compression algorithms. Off-the-shelf codecs can achieve relatively fast speed. Among neural codecs, TaCo-LL models achieve competitive compression performance with fewer parameters compared to P2LLM, Llama3-8B and RWKV-7B, and the encoding/decoding speed ranges from 317KB/s to 614KB/s.

## 4.3 LOSSY COMPRESSION FOR HUMAN VISION

**Evaluation Metrics.** We evaluate lossy compression performance using the Bjøntegaard Delta Rate (BD-Rate) (Bjontegaard, 2001) metric, which quantifies the average bitrate savings at a given level of distortion. A lower BD-Rate indicates better compression efficiency. We measure the reconstruction distortion using Peak Signal-to-Noise Ratio (PSNR) (Rosenfeld & Kak, 1982). The bitrate is assessed in bits per pixel (BPP), where uncompressed data corresponds to 24 BPP.

**Results.** Table. 4 benchmarks the lossy compression performance when using intra-frame compressors. Off-the-shelf intra-frame codecs like HM-Intra and VTM-Intra provide strong baselines, consistently delivering competitive performance. General-purpose codecs like JPEG2000 and JPEG-XL are included as standard baselines, but their performance is relatively poor. Neural compression methods pretrained on natural images, such as ELIC, LALIC, and TCM, show promising results on some datasets, but they fail to generalize to more structurally distinct data like ObjTac. In contrast, our TaCo-L model, trained on tactile datasets, achieves the best performance across all five datasets. It outperforms all baselines with BD-Rate reductions of -61.8% (TouchandGo), -24.3% (ObjectFolder), -27.4% (YCB-Slide), and -27.0% (ObjTac). Further, the force-based ObjTac dataset, which is derived from 3D force signals and mapped into RGB images, exhibits character-

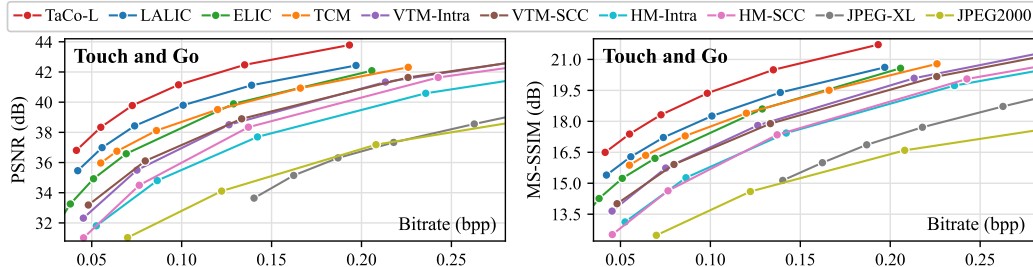

Figure 4: Rate-distortion curves on TouchandGo dataset, when applying intra-frame compression methods.

Table 6: Material classification results on TouchandGo, ObjectFolder-1.0 and object classification results on YCB-Slide. Best results are in **blue bold**, the second-best results are in bold, and the third-best in underline.

| | Compressor | BPP | SVM | Random Forest | K-NN | Linear Regression |
|---|---|---|---|---|---|---|
| **Touch and Go** | Uncompressed | 24 (1×) | **76.63%** | **74.88%** | **68.24%** | **73.51%** |
| | VTM-Intra | 0.213 | 74.08% | 72.53% | 65.06% | 70.87% |
| | JPEG-XL | 0.218 | 70.67% | 69.43% | 61.87% | 67.75% |
| | LALIC | 0.196 | 74.70% | 73.07% | 65.43% | 71.24% |
| | TaCo-L (ours) | 0.193 (124×) | **75.12%** | **73.55%** | **66.03%** | **71.89%** |
| **ObjectFolder** | Uncompressed | 24 (1×) | **44.11%** | **40.68%** | **37.14%** | **42.92%** |
| | VTM-Intra | 0.384 | 42.23% | 39.48% | 36.00% | 40.71% |
| | JPEG-XL | 0.499 | 40.27% | 37.63% | 34.36% | 38.74% |
| | LALIC | 0.477 | 41.00% | 38.28% | 35.44% | 39.91% |
| | TaCo-L (ours) | 0.453 (53×) | **43.08%** | **39.85%** | **36.27%** | **41.02%** |
| **YCB-Slide** | Uncompressed | 24 (1×) | **98.75%** | **98.72%** | **98.58%** | **99.18%** |
| | VTM-Intra | 0.118 | 97.36% | 96.41% | 97.08% | 97.24% |
| | JPEG-XL | 0.121 | 94.08% | 93.11% | 93.67% | 93.97% |
| | LALIC | 0.130 | 95.67% | 95.22% | 95.76% | 96.23% |
| | TaCo-L (ours) | 0.126 (190×) | **98.01%** | **97.35%** | **97.88%** | **98.20%** |

istics similar to screen content (large uniform regions and repetitive patterns). This makes screen-content-optimized codecs, VTM-SCC and HM-SCC, particularly effective on this dataset, achieving BD-Rates of -44.3% and -44.5%, respectively, when taking HM-Intra as the anchor. Table. 5 benchmarks the complexity of different lossy compression algorithms. For off-the-shelf codecs, the complexity increases along with the development of newer generations. For neural codecs, TaCo-L, adapted from the latest LALIC, achieves the best compression performance at the cost of incremental complexity, and the encoding/decoding FPS ranges from 4 FPS to 48 FPS at different resolutions. Fig. 4 presents a representative rate-distortion (RD) curve comparison on the TouchandGo dataset when using image compressors. TaCo-L achieves the SoTA performance across various bitrates.

## 4.4 LOSSY COMPRESSION FOR CLASSIFICATION

**Evaluation Metrics.** We evaluate the semantic fidelity of lossy compression using two tactile understanding tasks: material classification (on TouchandGo and ObjectFolder-1.0) and object classification (on YCB-Slide). For each dataset, we use four standard classifiers, SVM (Burges, 1998), Random Forest (Rigatti, 2017), K-NN (Peterson, 2009), and Linear Regression (Seber & Lee, 2012), with a fixed 60%/40% train-test data split. The top-1 accuracy is used as evaluation metric. Four representative lossy codecs, VTM-Intra, JPEG-XL, LALIC, and TaCo-L, are used for comparison.

**Results.** Table. 6 As shown in Table. 6, all methods achieve classification performance close to the uncompressed data, despite substantial bitrate savings (e.g., from 24 bpp to as low as 0.118 bpp). On TouchandGo, TaCo-L achieves 75.12% (SVM) and 71.89% (Linear Regression), similar to 76.63% and 73.51% when using uncompressed data. On ObjectFolder, where the task is more challenging, the top-1 accuracy under SVM drops slightly from 44.11% to 43.08% after compression with TaCo-L. On YCB-Slide, TaCo-L also preserves superior classification accuracy (98.01% and 98.20% when using SVM and Linear Regression, respectively), while reducing the bitrate by 190×.

Table 7: Evaluation results on the dexterous grasping. Best results are shown in **blue bold**, the second-best results are denoted in bold, and the third-best in underline. We also list the accuracy loss relative to the uncompressed data (8 bits/Byte) in parentheses.

| | Compressor | BPP | Small Obj. | Medium Obj. | Large Obj. | Deform. Obj. | Avg |
|---|---|---|---|---|---|---|---|
| $s_{\text{lift}}$ | Uncompressed | 24 | **54.7%** | **67.4%** | **69.2%** | **63.9%** | **63.8%** (-0.0%) |
| | JPEG-XL | 0.0505 | 47.2% | 58.0% | 59.7% | 55.1% | 55.0% (-8.8%) |
| | VTM-Intra | 0.0498 | **54.1%** | **66.6%** | 68.4% | **63.1%** | **63.1%** (-0.7%) |
| | LALIC | 0.0397 | 51.5% | 63.4% | 65.0% | 60.1% | 60.0% (-3.8%) |
| | TaCo-L (ours) | **0.0251** | 53.1% | 65.3% | **68.4%** | 61.9% | 62.2% (-1.6%) |
| $s_{\text{disturb}}$ | Uncompressed | 24 | **51.8%** | **65.8%** | **67.3%** | **61.8%** | **61.7%** (-0.0%) |
| | JPEG-XL | 0.0505 | 46.4% | 56.1% | 57.4% | 52.8% | 53.2% (-8.5%) |
| | VTM-Intra | 0.0498 | **52.5%** | **65.0%** | **66.7%** | **61.1%** | **61.3%** (-0.4%) |
| | LALIC | 0.0397 | 49.9% | 61.8% | 63.1% | 58.0% | 58.2% (-3.5%) |
| | TaCo-L (ours) | **0.0251** | 51.3% | 63.6% | 65.0% | 59.7% | 59.9% (-1.8%) |

## 4.5 LOSSY COMPRESSION FOR DEXTEROUS GRASPING

**Task Definition.** Many contact-rich manipulation algorithms for dexterous hands rely heavily on high-fidelity tactile signals, which motivates us to conduct dexterous grasping experiment. In real-world deployment scenarios, however, tactile data compression may affect the downstream performance of such algorithms. Therefore, we introduce this experiment to evaluate the impact of tactile compression quality on a realistic, task-driven benchmark. The goal of this task is to reach for an object, grasp and lift it. We build the simulation using Nvidia IssacSim Makoviychuk et al. (2021), and use a simple DexHand13 module Paxini. (2024) equipped with eleven tactile sensors. We modify the input tactile signals by compressing them first and then feed it into a tactile-aware reinforcement learning algorithm. In total we use 100 objects to evaluate the grasping performance, consisting of 29 small objects, 41 medium objects and 22 large objects, 8 deformable objects. For the following section, we list the performance for each category.

**Evaluation Metrics.** To ensure robust interference capabilities during grasping, we evaluate the performance using two evaluation metrics in the simulation. One is the success rate of lifting $s_{\text{lift}}$, recorded when objects maintain stability lifted to the height of 0.1m. The other is the success rate with disturbance resistance $s_{\text{disturb}}$, measured by applying 2.5N external forces along six axes for 2 s after lifting and the object moves below 0.02 m.

**Results.** Table. 7 benchmarks the grasping performance across different objects. Since the tactile signal simulated by Isaac Sim is relatively sparse, the achieved compression ratio is higher (up to 1000×) than what is typically attainable in the physical world (i.e. physical data achieve at most 22× (ObjTac) compression ratio). All compression methods successfully reduced the raw 24 bpp tactile signal to substantially smaller sizes, ranging from 0.025 bpp to 0.5 bpp, with only a moderate decrement in grasping success rate. Among them, TaCo-L outperforms JPEG-XL and LALIC, achieving a higher compression ratio while maintaining competitive lifting success rate of 62.2% compared to the baseline 63.8% and disturb-resistant grasping success rate of 59.9% compared to the baseline 61.7%. While our method TaCo-L is only approximately 1% of VTM's performance (62.2% vs 63.1% and 59.9% vs 61.3%) in terms of task success rate, it achieves significantly higher compression efficiency by operating at nearly half the bitrate (0.0251bpp vs. 0.0498 bpp).

## 5 CONCLUSION

This paper introduced the **TaCo** benchmark, the first comprehensive framework for evaluating tactile data codecs. This is a suite of 30 codecs, 5 datasets and 4 types of tasks to advance the research on tactile sensing and tactile data compression. Further, we presented **TaCo-LL** and **TaCo-L**, data-driven codecs that learn the latent distribution of tactile data end-to-end. Extensive experiments demonstrate that our proposed models establish a new state-of-the-art result, outperforming existing methods across lossless/lossy compression, classification, and grasping tasks. Our work provides a critical foundation and a baseline for future research in efficient tactile perception and transmission.

## ACKNOWLEDGMENT

This work was partly supported by the NSFC (62431015, 62571317, 62501387), the Fundamental Research Funds for the Central Universities, Shanghai Key Laboratory of Digital Media Processing and Transmission under Grant 22DZ2229005, 111 project BP0719010 and Okawa Research Grant.

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
