# TaCo: A Benchmark for Lossless and Lossy Codecs of Heterogeneous Tactile Data

**Zhengxue Cheng**[1]* **Yan Zhao**[1] **Keyu Wang**[1] **Hengdi Zhang**[2] **Li Song**[1]
[1] Shanghai Jiao Tong University, Shanghai, China
[2] Paxini Tech., Shenzhen, China
zxcheng@sjtu.edu.cn

## A  Appendix

### A.1  Baseline and Implementation Details

We benchmark the performance of tactile compression on five representative tactile datasets: Touch and Go[1], ObjectFolder 1.0[2], SSVTP[3], YCB-Slide[4], and ObjTac[5], as detailed in Section 3.1. Specifically, 70% of the samples from Touch and Go and ObjectFolder are used for training, while the remaining 30%, along with the full SSVTP and ObjTac datasets, are used for evaluation. For the Touch and Go dataset, while the official guideline recommends splitting by collection trajectories, it does not specify an exact train/test ratio or content. We followed this recommendation by grouping data at the trajectory level and applied a common 70% and 30% split for training and testing. Each trajectory was then decomposed into individual frames, ensuring that all frames from the same trajectory are contained entirely within either the training or the testing set, avoiding any data leakage.

For **TaCo-LL**, adapted from DualComp-I (Zhao et al., 2025), we train it using the FusedAdam optimizer (NVIDIA, 2018) with a cosine annealing learning rate schedule. (Loshchilov & Hutter, 2016), starting from $1 \times 10^{-4}$ and decaying to $2 \times 10^{-5}$ over 20 epochs. The batch size is set to 64. The model is trained with a standard cross-entropy loss:

$$\mathcal{L}_{\text{TaCo-LL}} = -\sum q \log p \tag{1}$$

where $q$ and $p$ are the target and predicted distributions, respectively.

For **TaCo-L**, We train our models using the Adam optimizer (Kingma & Ba, 2014) with a batch size of 8. The model is optimized with a rate-distortion loss:

$$\mathcal{L}_{\text{TaCo-L}} = R + \lambda \cdot \text{MSE} \tag{2}$$

where $R$ denotes the estimated bitrate and $\lambda$ controls the trade-off between rate and distortion. For MSE-optimized models, $\lambda$ is set to $\{0.0018, 0.0067, 0.025, 0.0483\}$ to achieve different bitrates. The learning rate is set to $1 \times 10^{-4}$ for 40 epochs, and then decayed to $1 \times 10^{-5}$ for another 4 epochs. During training, input tactile images are randomly cropped or padded to $256 \times 256$ resolution.

These two models are trained using two NVIDIA A100 GPUs with mixed precision enabled.

### A.2  Cross-Dataset Compression Performance

Furthermore, the above training datasets are mainly collected on rigid and lambertian objects, and we also validate compression performance on two new test datasets: Active Cloth (Yuan et al., 2018) covering soft and textured objects, and ObjectFolder-2.0 comprising a wide variety of everyday 3D objects, as shown in Table. 1. Due to the large scale of both datasets, we conduct quick validation on approximately the first 10% of the data from each: 10 objects from Active Cloth and 100 objects

---

*Corresponding Author
[1] https://touch-and-go.github.io/
[2] https://objectfolder.stanford.edu/
[3] https://sites.google.com/berkeley.edu/ssvtp
[4] https://github.com/rpl-cmu/YCB-Slide
[5] https://readerek.github.io/Objtac.github.io/

Table 2: Comparison of lossless compression performance (bits/Byte) on two additional tactile datasets to validate the cross-dataset performance. The best results are highlighted in **bold blue**, second-best in **bold**, and third in underline. For TaCo-LL, 12M/48M/96M denotes the model parameter. To show the compression performance more clearly, we also list the compression ratios relative to the uncompressed data (8 bits/Byte) in parentheses only for top three results.

| | Compressor | bits/Byte↓ | |
| --- | --- | --- | --- |
| | | ActiveCloth | ObjectFolder-2.0 |
| | uncompressed | 8 (1×) | 8 (1×) |
| **Off-the-Shelf** | gzip (Pasco., 1996) | 2.762 | 4.040 |
| | zstd (Meta., 2015) | 2.771 | 4.037 |
| | bzip2 (Seward, 2000) | 2.771 | 4.103 |
| | FLIF (Sneyers, 2015) | 0.882 | 3.831 |
| | BPG (Bellard, 2014) | 1.645 | 3.792 |
| | WebP (Google, 2010) | 1.063 | 3.676 |
| | JPEG-XL (Team, 2021) | 0.841 (9.5×) | 3.659 |
| | JPEG2000 (ISO/IEC, 2000) | 1.804 | 4.061 |
| | PNG (Boutell, 1997) | 2.667 | 4.035 |
| **Neural** | DLPR (Bai et al., 2024) | 1.453 | 3.852 |
| | P2LLM (Chen et al., 2024) | 2.193 | 3.470 |
| | Llama3* (Deletang et al., 2024) | 2.620 | 3.659 |
| | RWKV* (Deletang et al., 2024) | 2.640 | 3.800 |
| | DualComp-I (Zhao et al., 2025) | 1.158 | 3.308 |
| | TaCo-LL-12M (ours) | 1.059 | 3.179 (2.5×) |
| | TaCo-LL-48M (ours) | **0.816** (10×) | **3.002** (2.7×) |
| | TaCo-LL-96M (ours) | **0.723** (11×) | **2.855** (2.8×) |

from ObjectFolder-2.0. Lossless and lossy compression are performed, as summarized in Table 2 and Table 3.

Table 1: Introduction of two additional tactile datasets, where ActiveCloth (Yuan et al., 2018) consists of 153 varied pieces of clothes and ObjectFolder-2.0 (Gao et al., 2022) mainly extends ObjectFolder-1.0 (Gao et al., 2021) with 100 virtualized objects to 1000 common household real objects.

| Dataset | #Objects | #Frames | Resolution | Sensor |
| --- | --- | --- | --- | --- |
| ActiveCloth (Yuan et al., 2018) | 153 | 494655 | $640 \times 480 \times 30Hz$ | GelSight (Yuan et al., 2017) |
| ObjectFolder 2.0 (Gao et al., 2022) | 1000 | 76000 | $120 \times 160 \times 30Hz$ | GelSight (Yuan et al., 2017) |

When comparing Active Cloth and Touch and Go at the same resolution of $640 \times 480$, our TaCo-LL model, with 96M parameters, achieves the best performance on both datasets. It achieves 0.723 bit/Byte (in Table 2) and 0.447 bit/Byte (in Table 2), corresponding to compression ratios of $11\times$ and $18\times$, respectively. The results also suggest that soft objects in Active Cloth are more difficult to compress than rigid objects, as deformable surfaces tend to generate more complex tactile data. When comparing ObjectFolder-1.0 and ObjectFolder-2.0 at the same resolution of $120 \times 160$, all the compression methods basically achieve consistent results and our TaCo-LL also achieve the best performance with 2.855 bit/Byte, corresponding to compression ratios of $2.8\times$.

These findings are further supported by the BD-Rate comparisons in Table 3, where TaCo-L consistently achieves the lowest BD-Rate across both ActiveCloth and ObjectFolder-2.0 datasets, outperforming state-of-the-art neural compressors such as ELIC, LALIC, and TCM.

## A.3 CROSS-OBJECT COMPRESSION PERFORMANCE

Table. 4 and Table. 5 present a evaluation of cross-object lossless compression performance (in bits/Byte). Table. 4 focuses on rigid objects from TouchandGo and ObjTac datasets, while Table. 5 evaluates soft objects from ActiveCloth and ObjTac datasets. A key observation is that compression performance is influenced primarily by the type of sensor modality, as evidenced by consistent trends within datasets from the same source (e.g., ActiveCloth vs. TouchandGo). However, the physical properties of the object (like rigidity or softness) also have an obvious impact.

Table 3: Lossy compression performance (BD-Rate) on two additional tactile datasets to validate the cross-dataset performance. The best results are shown in **blue bold**, the second-best in **bold**, and the third-best in underline. For TaCo, 12M/48M/96M denotes the model parameter.

| | Compressor | BD-Rate (%)↓ | |
|---|---|---|---|
| | | ActiveCloth | ObjectFolder-2.0 |
| **Off-the-Shelf** | HM-Intra (Sullivan et al., 2012) | 0% | 0% |
| | HM-SCC (Sullivan et al., 2012) | -12.9% | 2.2% |
| | VTM-Intra (Bross et al., 2021) | -28.0% | **-21.0%** |
| | VTM-SCC (Bross et al., 2021) | -26.1% | -19.3% |
| | JPEG-XL (Team, 2021) | 46.9% | 80.7% |
| | JPEG2000 (ISO/IEC, 2000) | 86.5% | 79.0% |
| **Neural** | ELIC (He et al., 2022) | **-57.1%** | 3.2% |
| | LALIC (Feng et al., 2025) | -54.8% | 2.8% |
| | TCM (Liu et al., 2023) | -49.8% | 23.7% |
| | TaCo-L (Ours) | **-65.4%** | **-26.4%** |

Table 4: Cross-object lossless compression performance (bits/Byte) on RIGID objects. The best results are highlighted in **bold blue**, the second-best in **bold**, and the third in underline. In TouchandGo, Tree and Wood are training objects, while Concrete is unseen. The three objects in ObjTac are all unseen objects (*).

| | Compressor | bits/Byte↓ | | | | | |
|---|---|---|---|---|---|---|---|
| | | **Touch and Go** | | | **ObjTac** | | |
| | | Tree | Wood | Concrete* | Stone* | Pebble* | Tile* |
| | uncompressed | 8 (1×) | 8 (1×) | 8 (1×) | 8 (1×) | 8 (1×) | 8 (1×) |
| **Off-the-Shelf** | gzip (Pasco., 1996) | 2.531 | 2.082 | 2.214 | 1.100 | 0.943 | 0.529 |
| | zstd (Meta., 2015) | 2.327 | 2.033 | 2.080 | 1.098 | 0.930 | 0.505 |
| | bzip2 (Seward, 2000) | 2.486 | 2.068 | 2.186 | 1.123 | 0.976 | 0.541 |
| | FLIF (Sneyers, 2015) | 0.865 | 0.737 | 0.782 | 0.697 | 0.648 | 0.303 |
| | BPG (Bellard, 2014) | 1.395 | 1.141 | 1.246 | 1.061 | 0.847 | 0.453 |
| | WebP (Google, 2010) | 1.000 | 0.855 | 0.924 | 0.848 | 0.720 | 0.387 |
| | JPEG-XL (Team, 2021) | 0.796 | **0.670** | **0.730** | 0.742 | 0.656 | 0.372 |
| | JPEG2000 (ISO/IEC, 2000) | 1.617 | 1.421 | 1.540 | 1.181 | 0.990 | 0.500 |
| | PNG (Boutell, 1997) | 2.765 | 2.249 | 2.390 | 1.097 | 0.939 | 0.527 |
| **Neural** | DLPR (Bai et al., 2024) | 1.127 | 0.935 | 1.062 | 0.906 | 0.917 | 0.551 |
| | P2LLM (Chen et al., 2024) | 1.946 | 1.475 | 1.832 | 0.933 | 0.911 | 0.534 |
| | Llama3* (Deletang et al., 2024) | 2.479 | 2.010 | 2.145 | 1.098 | 1.102 | 0.809 |
| | RWKV* (Deletang et al., 2024) | 2.558 | 2.120 | 2.396 | 1.175 | 1.110 | 0.832 |
| | DualComp-I (Zhao et al., 2025) | 0.840 | 0.726 | 0.857 | 0.810 | 0.685 | 0.339 |
| | TaCo-LL-12M (ours) | 0.810 | 0.704 | 0.815 | 0.796 | 0.680 | 0.336 |
| | TaCo-LL-48M (ours) | **0.719** | **0.611** | **0.730** | **0.635** | **0.627** | **0.300** |
| | TaCo-LL-96M (ours) | **0.607** | **0.598** | **0.700** | **0.590** | **0.596** | **0.288** |

## A.4  ANALYSIS OF TACTILE DATA CHARACTERISTIC

Fig. 1 includes samples from YCB-Slide (Digit sensor, rigid objects such as Sugar Box, Mug, and Mustard Bottle), SSVTP (Digit sensor, soft cloth objects like Cloth Corner and Interior), Active Cloth (GelSight sensor, soft fabrics including Cloth-12, 29, and 33), Touch and Go (GelSight sensor, rigid surfaces such as Tree, Wood, and Concrete), and ObjTac (Force sensor, both soft objects like Jeans, Leather Bag, Sponge, and rigid ones like Pebble, Stone, Tile). As depicted, the entropy maps and FFT spectra show that tactile images are dominated by low-frequency energy with highly repetitive, grid-like spatial structures. These signals exhibit strong directional patterns and locally predictable regions, leading to sparse residuals after prediction. As a result, block-based lossy codecs such as SCC perform especially well, since their intra prediction, palette modes, and transform coding are optimized for smooth, structured, and repetitive content. The same properties also explain the behavior of lossless codecs: low entropy regions compress extremely well, while periodic patterns favor context-based or LZ-type entropy models.

## A.5  MORE LOSSY COMPRESSION PERFORMANCE FOR HUMAN VISION

In addition to intra-frame compression methods, Table. 6 benchmarks the use of video codecs for compressing tactile data, focusing on their ability to exploit inter-frame redundancy. It can be seen

Table 5: Cross-object lossless compression performance (bits/Byte) on SOFT objects. The best results are highlighted in **bold blue**, the second-best in **bold**, and the third in underline. All these objects are unseen (*).

| | Compressor | bits/Byte↓ | | | | | |
|---|---|---|---|---|---|---|---|
| | | **Active Cloth** | | | **ObjTac** | | |
| | | Cloth-12* | Cloth-29* | Cloth-33* | Sponge* | Jeans* | Leather Bag* |
| | uncompressed | 8 (1×) | 8 (1×) | 8 (1×) | 8 (1×) | 8 (1×) | 8 (1×) |
| Off-the-Shelf | gzip (Pasco., 1996) | 3.540 | 1.902 | 3.609 | 0.214 | 0.178 | 0.129 |
| | zstd (Meta., 2015) | 3.550 | 1.911 | 3.619 | 0.210 | 0.173 | 0.128 |
| | bzip2 (Seward, 2000) | 3.552 | 1.907 | 3.619 | 0.252 | 0.206 | 0.144 |
| | FLIF (Sneyers, 2015) | 1.097 | **0.668** | **1.101** | 0.106 | 0.079 | **0.071** |
| | BPG (Bellard, 2014) | 2.043 | 1.194 | 2.064 | 0.207 | 0.148 | 0.144 |
| | WebP (Google, 2010) | 1.319 | 0.802 | 1.323 | 0.178 | 0.092 | **0.065** |
| | JPEG-XL (Team, 2021) | **1.055** | **0.621** | **1.057** | 0.106 | **0.076** | 0.088 |
| | JPEG2000 (ISO/IEC, 2000) | 2.143 | 1.419 | 2.140 | 0.246 | 0.263 | 0.204 |
| | PNG (Boutell, 1997) | 3.549 | 1.586 | 3.619 | 0.213 | 0.197 | 0.165 |
| Neural | DLPR (Bai et al., 2024) | 1.877 | 1.590 | 1.985 | 0.148 | 0.094 | 0.150 |
| | P2LLM (Chen et al., 2024) | 2.033 | 1.724 | 2.082 | 0.170 | 0.142 | 0.143 |
| | Llama3* (Deletang et al., 2024) | 3.147 | 1.883 | 3.251 | 0.492 | 0.185 | 0.158 |
| | RWKV* (Deletang et al., 2024) | 3.219 | 1.890 | 3.238 | 0.510 | 0.179 | 0.163 |
| | DualComp-I (Zhao et al., 2025) | 1.696 | 1.125 | 2.000 | 0.105 | 0.109 | 0.110 |
| | TaCo-LL-12M (ours) | 1.710 | 1.147 | 1.991 | 0.106 | 0.100 | 0.113 |
| | TaCo-LL-48M (ours) | 1.332 | 0.877 | 1.930 | **0.100** | 0.078 | 0.095 |
| | TaCo-LL-96M (ours) | **1.016** | 0.725 | 1.255 | **0.097** | **0.053** | 0.079 |

that DCVC-RT achieves the best performance on most dataset, followed by DCVC-FM and VVenC. Since each row of an ObjTac image corresponds to a distinct timestamp, the dataset does not have video format and video compression evaluation.

Table 6: Evaluation on lossy compression performance with regard to intra-frame and inter-frame correlations. The best results are denoted in **bold**, and the second-best in underline

| | Compressor | BD-Rate (%)↓ | | | |
|---|---|---|---|---|---|
| | | TouchandGo | ObjectFolder | SSVTP | YCB-Slide |
| Off-the-Shelf | x265 (Sullivan et al., 2012) | 0% | 0% | 0% | 0% |
| | SVT-AV1 (Han et al., 2021) | -40.6% | -34.2% | -28.2% | -12.1% |
| | VVenC (Bross et al., 2021) | -67.6% | -16.4% | -33.6% | -52.2% |
| Neural | DCVC-DC (Li et al., 2023) | -75.6% | -12.2 % | -20.4% | -27.1% |
| | DCVC-FM (Li et al., 2024) | **-80.0%** | -43.8% | -45.2% | -58.1% |
| | DCVC-RT (Jia et al., 2025) | -78.1% | **-48.8%** | **-50.9%** | **-65.5%** |

To complement the BD-Rate results, we present full rate-distortion (RD) curves for each dataset in both image-based (Fig. 2) and video-based (Fig. 3) compression settings, using PSNR and MS-SSIM as distortion metrics. For the TouchandGo dataset, image-based RD curves are shown in the main paper.

These curves provide a more detailed view of compression performance across bitrates. In the image-based setting, JPEG-XL and JPEG2000 consistently result to relatively poor performance. TaCo-L consistently achieves the best performance across all datasets except ObjTac, where screen-content-coding (SCC) modes in VTM and HM are particularly effective due to the screen-content-like patterns. In the video-based setting, neural codecs like DCVC variants outperform traditional video codecs like x265, SVT-AV1 and VVenC, especially in the low-bitrate region.

Aside from objective metrics, We also use the YCB-Slide dataset as an example and provide per-pixel RMSE error maps of the reconstructed tactile signals in Fig. 4. As discussed in Fig.1 of the main paper, although tactile images carry meaningful physical information, their visual appearance is often unintuitive for human interpretation. Therefore, instead of relying on perceptual inspection, we quantify local reconstruction discrepancies using the pixel-wise RMSE,

$$\text{RMSE}(x, \hat{x}) = \sqrt{(x - \hat{x})^2}. \tag{3}$$

As shown in Fig. 4, all methods produce relatively low reconstruction errors even at low bitrates (below 0.1 bpp, i.e., over $240\times$ compression), indicating that lossy compression at relatively high ratios is generally acceptable for human-viewing purposes.

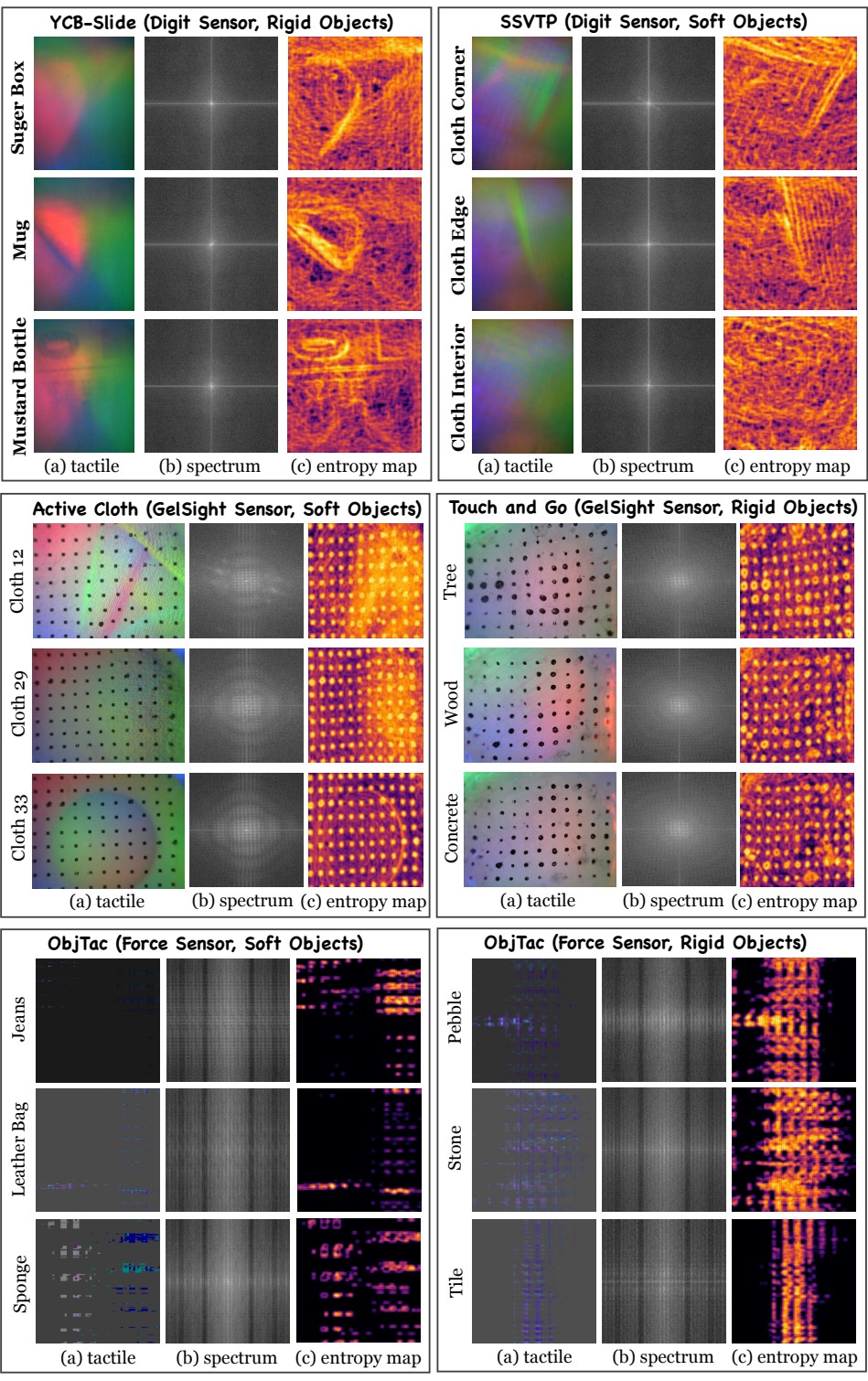

Figure 1: Visualization of tactile data characteristics across different datasets, sensors, and object types. Each subfigure displays the raw tactile image, its frequency spectrum, and the corresponding entropy map.

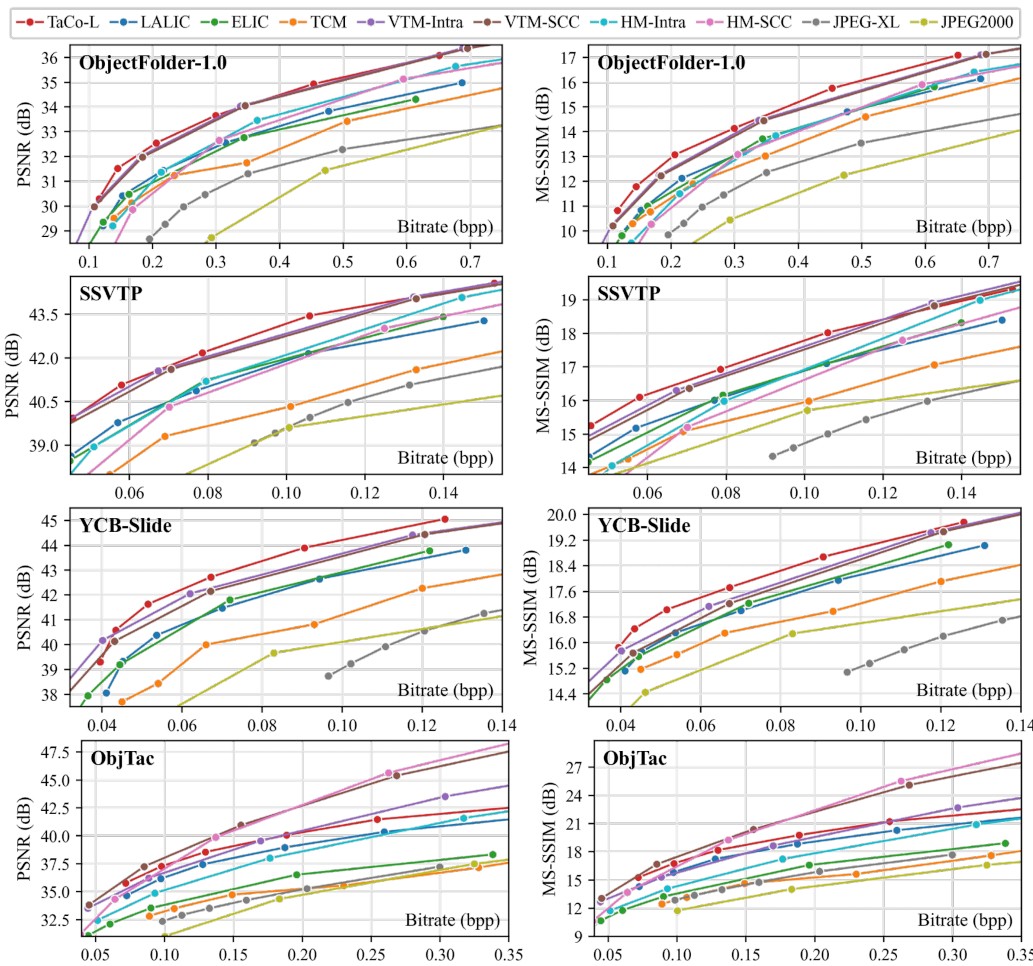

Figure 2: Rate-distortion performance across four tactile datasets when treating tactile data as images.

## A.6 MORE LOSSY COMPRESSION RESULTS FOR CLASSIFICATION

To complement the results in the main paper and illustrate the full-bitrate performance, we present the bitrate-accuracy curves on Touch and Go, ObjectFolder-1.0, and YCB-Slides dataset, as shown in Fig. 5. Each curve shows how classification accuracy changes as the bitrate varies, with dotted lines indicating the performance on uncompressed data (24 bpp). These plots provide a more comprehensive view of semantic preservation across different compression levels.

Specifically, the bitrate is varied by adjusting the quantization parameter (QP) for each compressor. For each classification task, we split the reconstructed data into 60% for training and 40% for testing, and apply four standard classifiers (SVM, Random Forest, K-NN, and Linear Regression) to perform material classification (TouchandGo and ObjectFolder) or object classification (YCB-Slide). Overall, even at over $200\times$ compression, the impact on classification accuracy remains minor for all methods, suggesting that lossy compression can be applied without substantially compromising downstream understanding tasks. Among them, TaCo-L consistently achieves the highest accuracy across the full bitrate range, and closely approaches the accuracy of raw data (24 bpp).

## A.7 MORE LOSSY COMPRESSION RESULTS FOR DEXTEROUS GRASPING

We further visualize the simulation environment from the IssacSim and part of the assets, as shown in Fig. 6. The hand model is based on the Paxini DexHand13 Paxini. (2024), which has four fingers and a total of 16 DoF. Each finger except the thumb is equipped with three tactile sensors and the

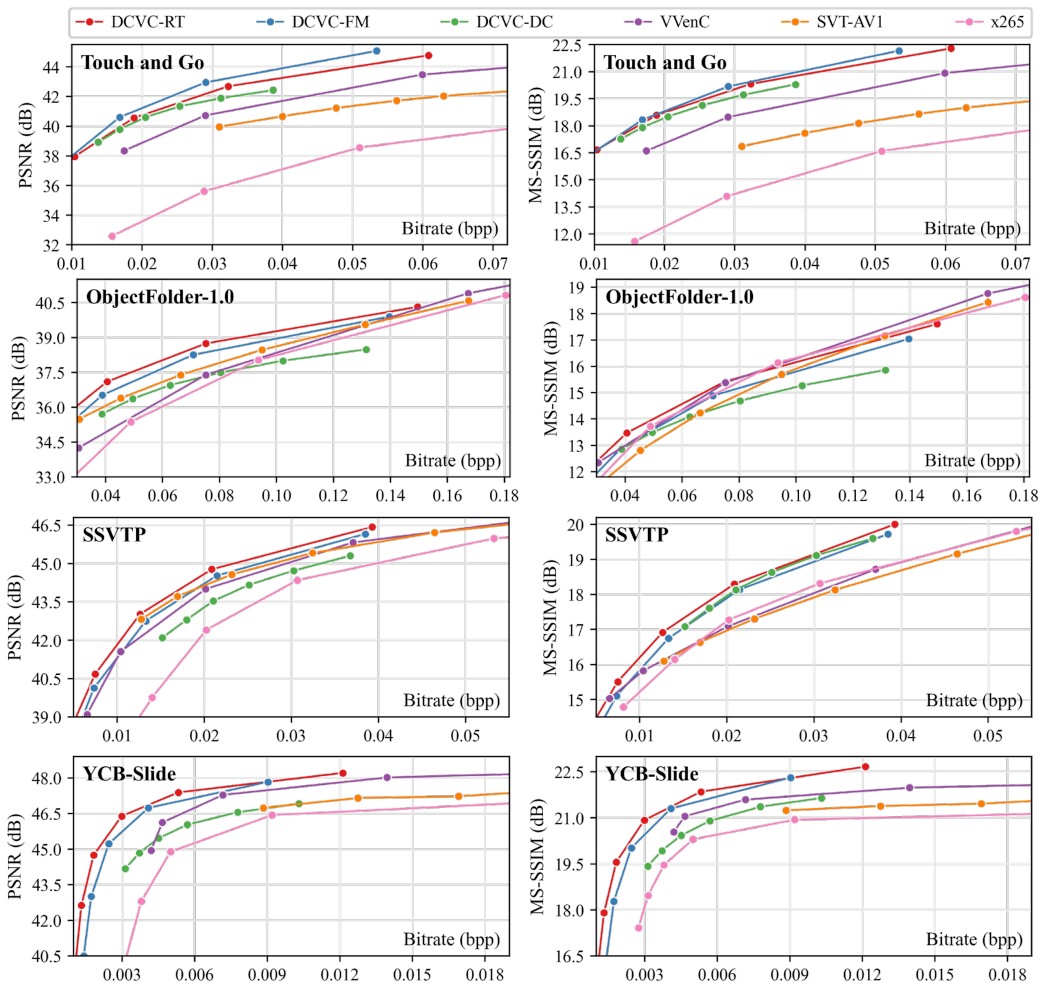

Figure 3: Rate-distortion performance across four tactile datasets when treating tactile data as videos.

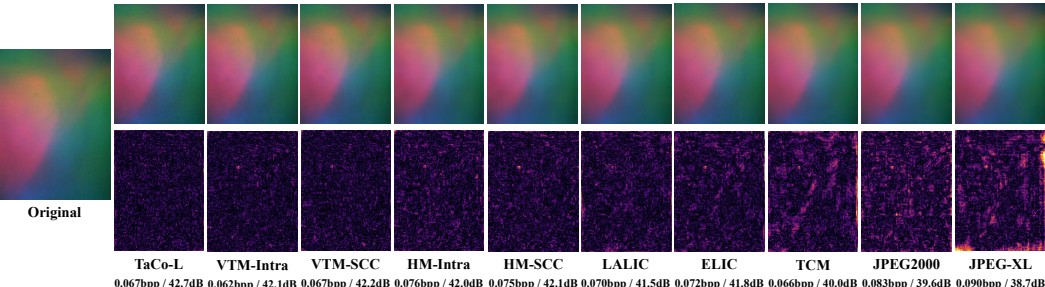

Figure 4: Visualization of reconstructed tactile images (top row) and their corresponding per-pixel root mean squared error (RMSE) maps (bottom row) on the YCB-Slide dataset. The RMSE maps highlight local reconstruction errors, with brighter regions indicating larger residuals.

thumb finger is equipped with two tactile sensors, resulting to a total of 11 tactile sensors. We deploy a simple asymmetric actor-critic (AAC) network with the tactile data as the input, to learn the dexterous grasping for general objects (Wang et al., 2025). Although the grasping success rate of our baseline model is not very high, we focus on the impact of tactile compression.

We have conducted grasping experiments in real-world settings and employed four mature encoders (JPEG2000, JPEG XL, BPG, VTM) to compress tactile signals with varying quantization parameters

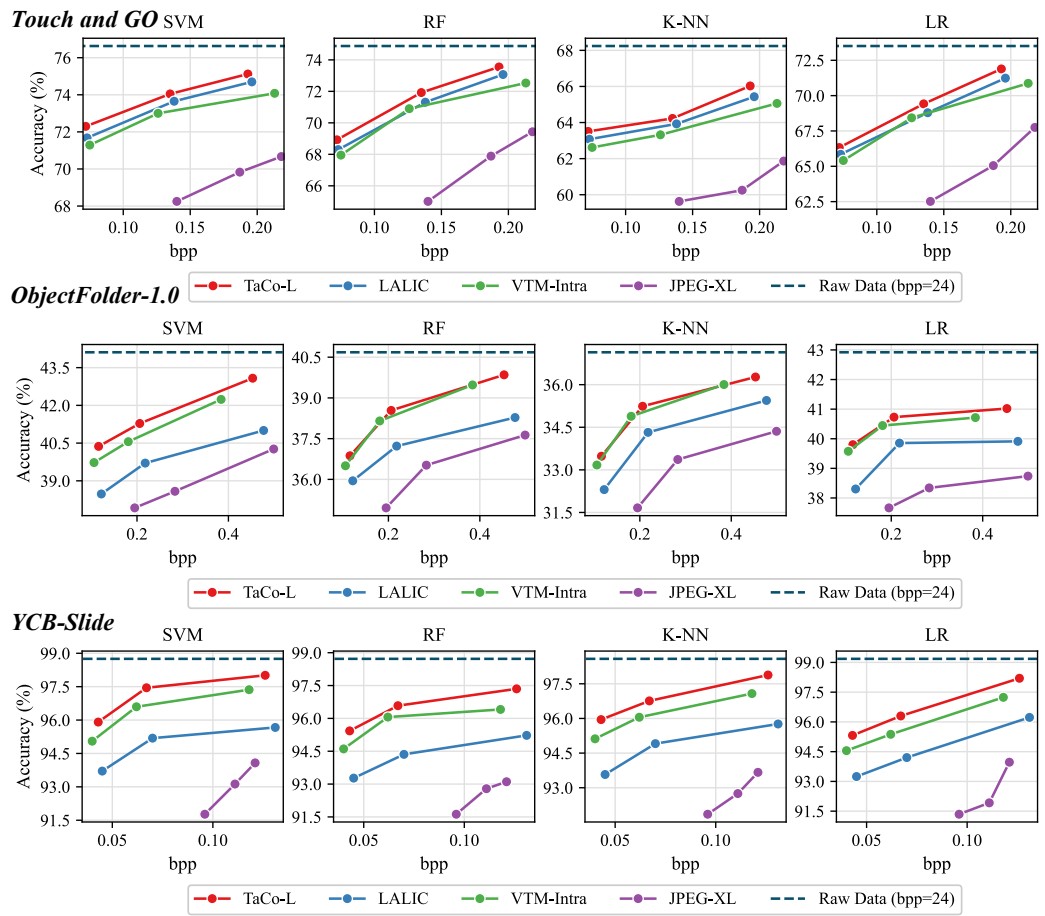

Figure 5: Bpp-accuracy curves for material classification task on the TouchandGo and ObjectFolder-1.0 datasets, and object classification task on the YCB-Slide dataset.

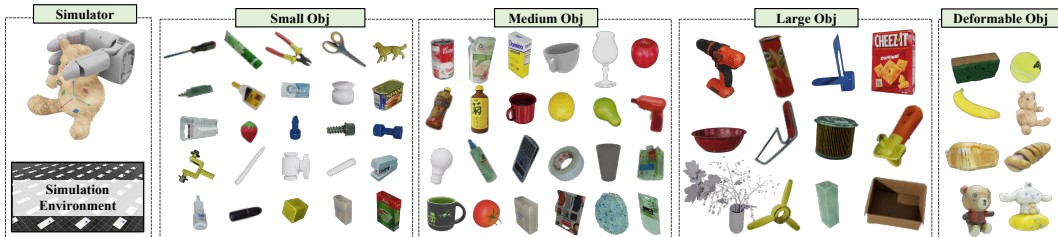

Figure 6: Simulation environment and part of object assets we use in the grasping exeriments.

(QP). Using four fingertip positions as primary observation metrics, we present the sensory force data along the x, y, and z axes across these four fingertips, with the results illustrated in Fig. 7.

Fig. 8, Fig. 9, Fig. 10 and Fig. 11 illustrate the visualization results of tactile signals from four fingertips using four different codecs in real-world experiments. Meanwhile, Fig. 12, Fig. 13, Fig. 14 and Fig. 15 illustrate the visualization results of tactile signals from four fingertips using four different codecs in the simulations. These figures demonstrate that the compression algorithm itself does not actually affect the main variation distribution of the tactile data, and therefore will not have a catastrophic impact on the accuracy of real-world tasks.

Regarding the implementation details, the simulation environment for the reinforcement learning controller operates at a control frequency of 100 Hz, which is determined by the simulation time

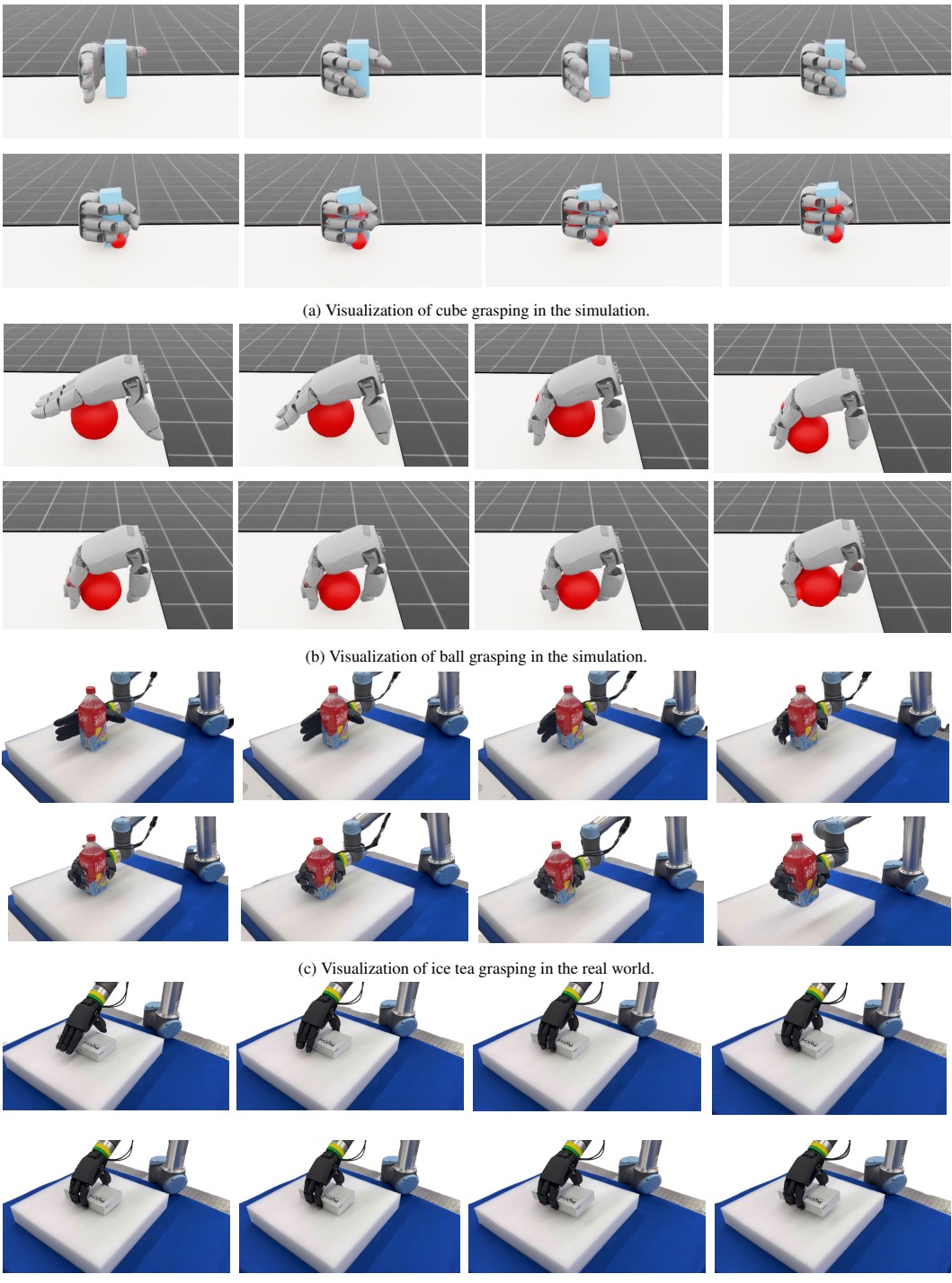

(a) Visualization of cube grasping in the simulation.

(b) Visualization of ball grasping in the simulation.

(c) Visualization of ice tea grasping in the real world.

(d) Visualization of box grasping in the real world.

Figure 7: Visualization of grasping sequences in the simulation and real-world experiments.

step of 0.01 seconds. Specifically, (1) the tactile sensors are updated at every simulation step, resulting in a tactile sampling rate of 100 Hz. (2) The overall latency of the control loop is approximately 0.01 seconds, plus the time required for policy inference. The policy inference is performed using an ONNX model, and the inference time is logged during execution. If the inference time exceeds the simulation time step, the control frequency may decrease, and the latency would increase ac-

cordingly. (3) When the combined codec and inference latency approximately equals the simulation update interval, the additional delay introduced to the simulation environment becomes negligible, as it aligns with the natural timing cycle of the control loop. However, in the current implementation, the control command is applied in the same simulation step after inference, so the latency is primarily determined by the simulation step and the inference time.

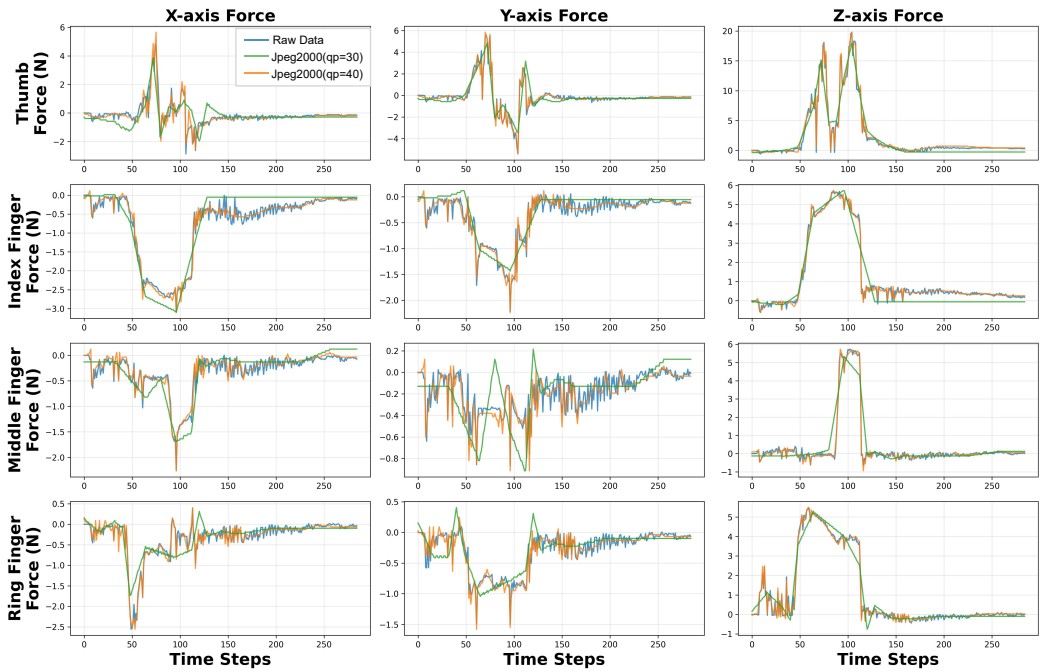

Figure 8: Visualization of tactile signals in the real-world experiments with JPEG2000 as the codec.

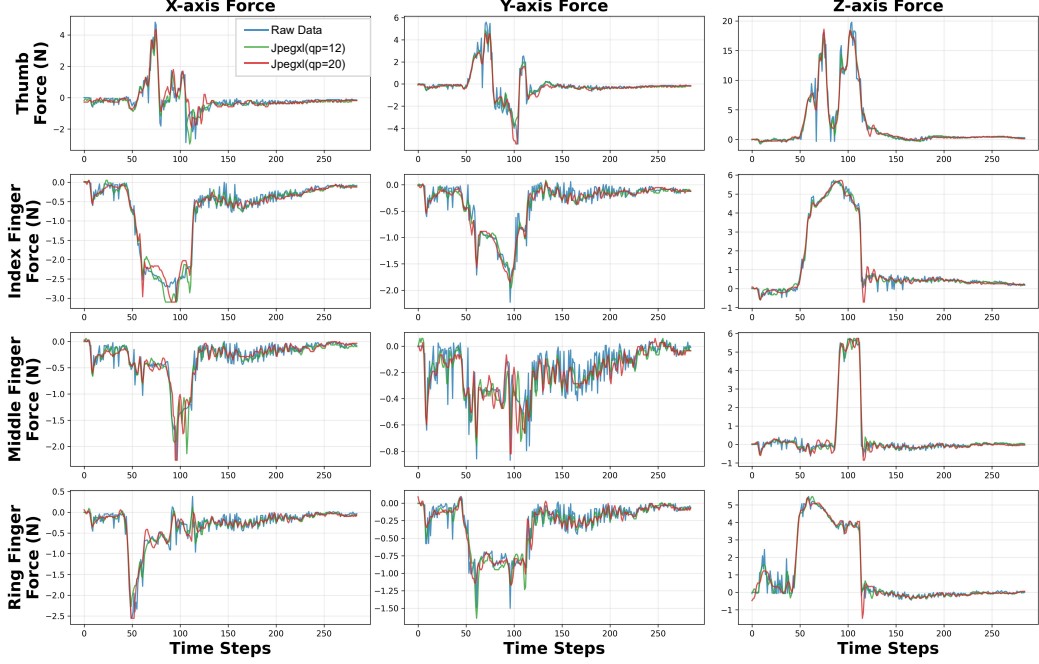

Figure 9: Visualization of tactile signals in the real-world experiments with JPEG-XL as the codec.

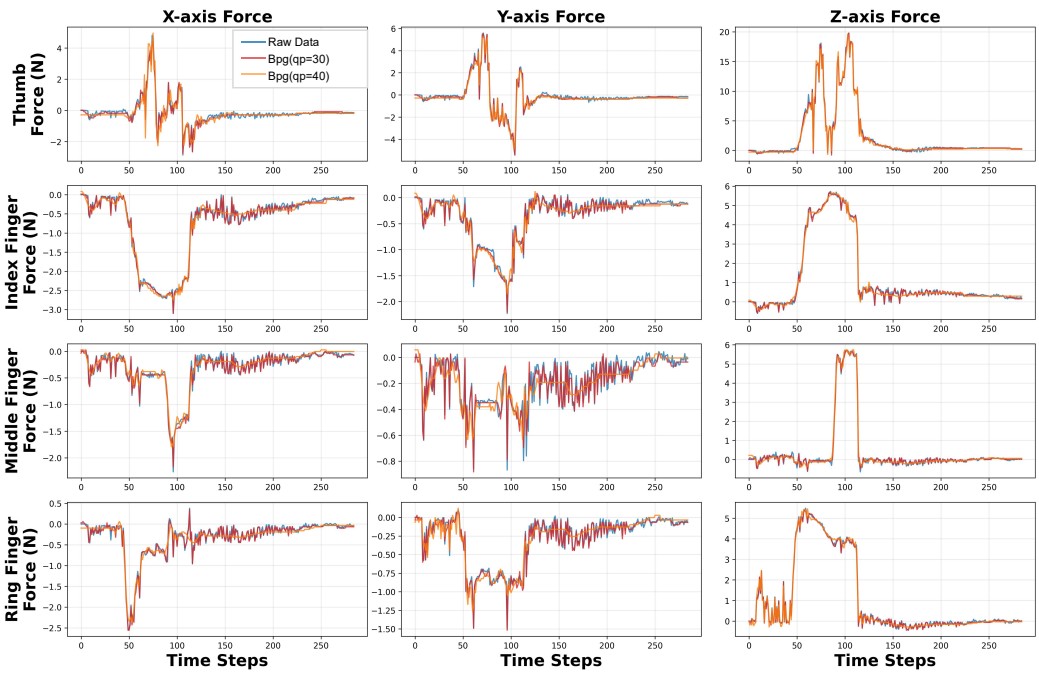

Figure 10: Visualization of tactile signals in the real-world experiments with BPG as the codec.

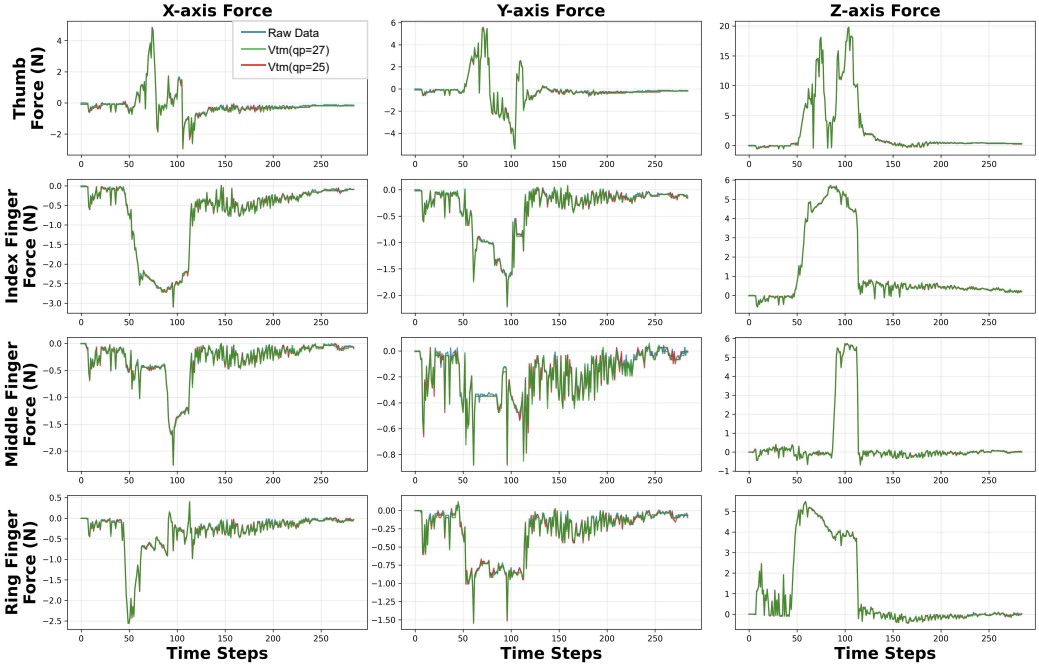

Figure 11: Visualization of tactile signals in the real-world experiments with VTM as the codec.

## A.8 OUR MOTIVATION AND FUTURE WORK

In this section, we simply present the need for advancing tactile compression. The development of this tactile codec benchmark is motivated by three critical challenges in practical robotics applications. First, for dexterous manipulation, tactile data from high-resolution sensor arrays on robot hands can consume a significant portion of the available bandwidth. This is analogous to the Cortical

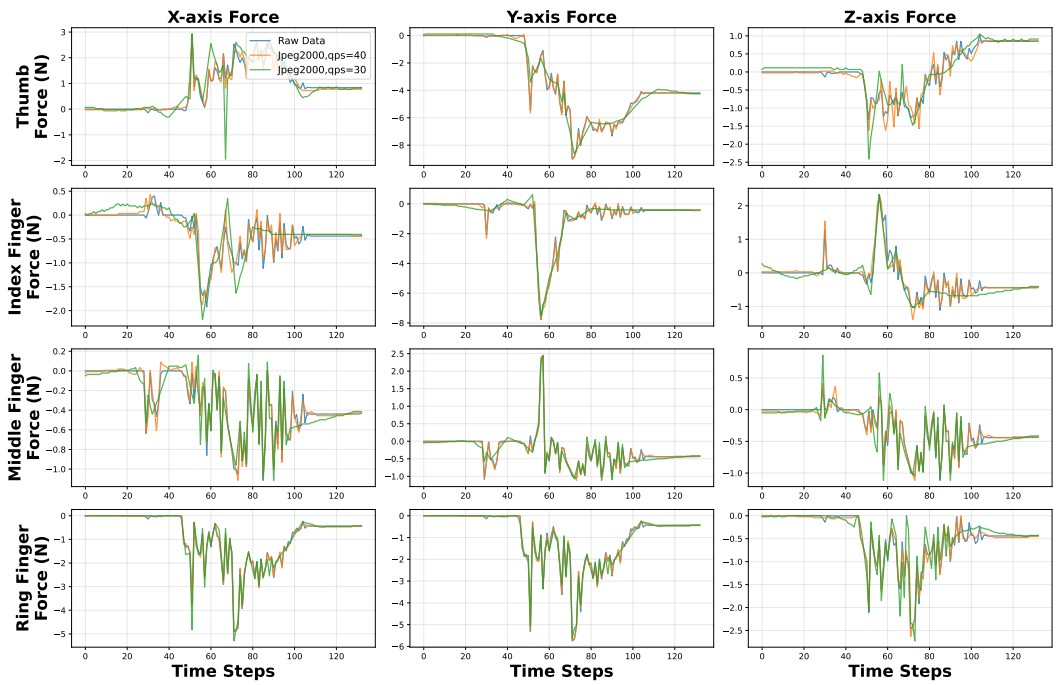

Figure 12: Visualization of tactile signals in the simulation experiments with JPEG2000 as the codec.

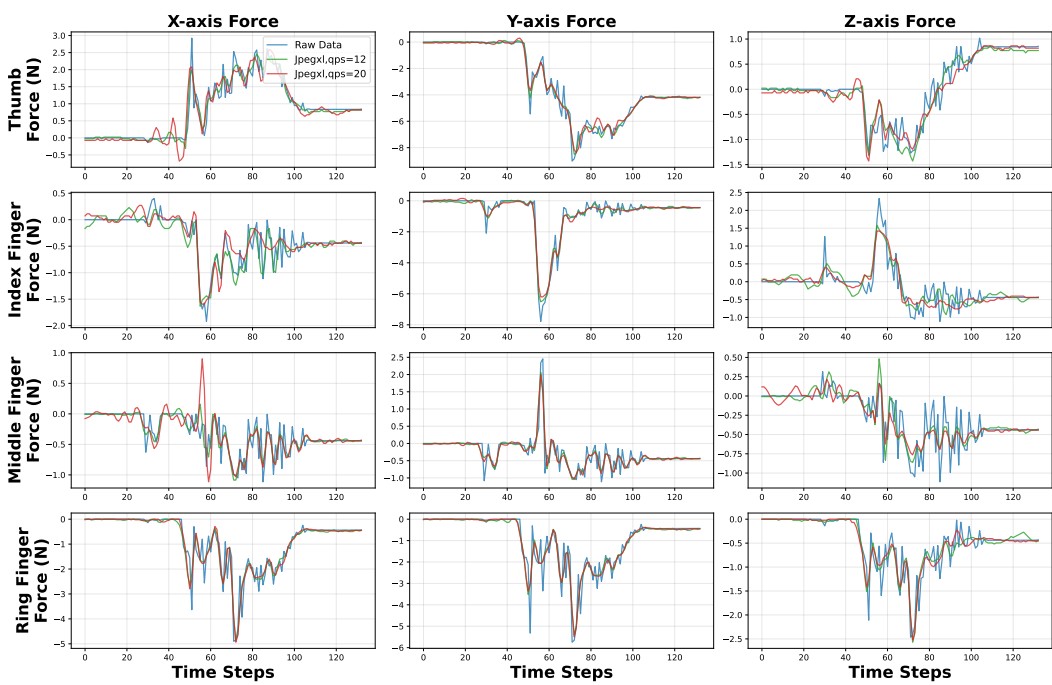

Figure 13: Visualization of tactile signals in the simulation experiments with JPEG-XL as the codec.

Homunculus, where the hands claim a disproportionately large share of neural resources. The limited bandwidth of low-cost microcontrollers (MCUs) embedded in such hands creates a fundamental bottleneck for real-time sensorimotor control.

Second, in robotic tele-operation systems, achieving stable and transparent remote control requires low-latency, high-fidelity transmission of tactile signals. Effective compression is paramount to

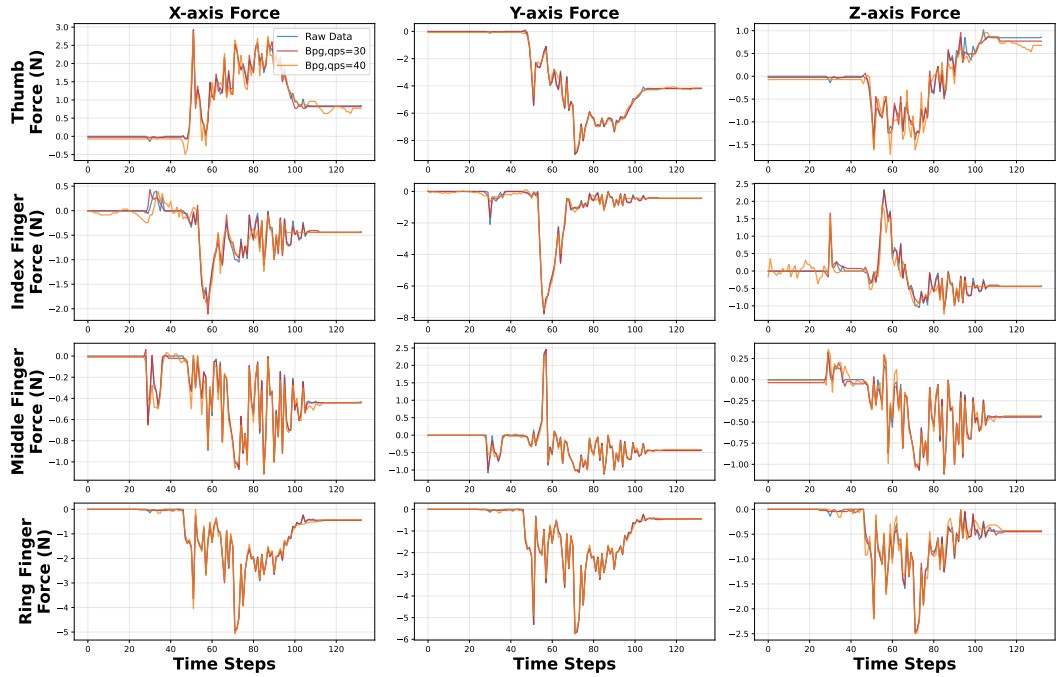

Figure 14: Visualization of tactile signals in the simulation experiments with BPG as the codec.

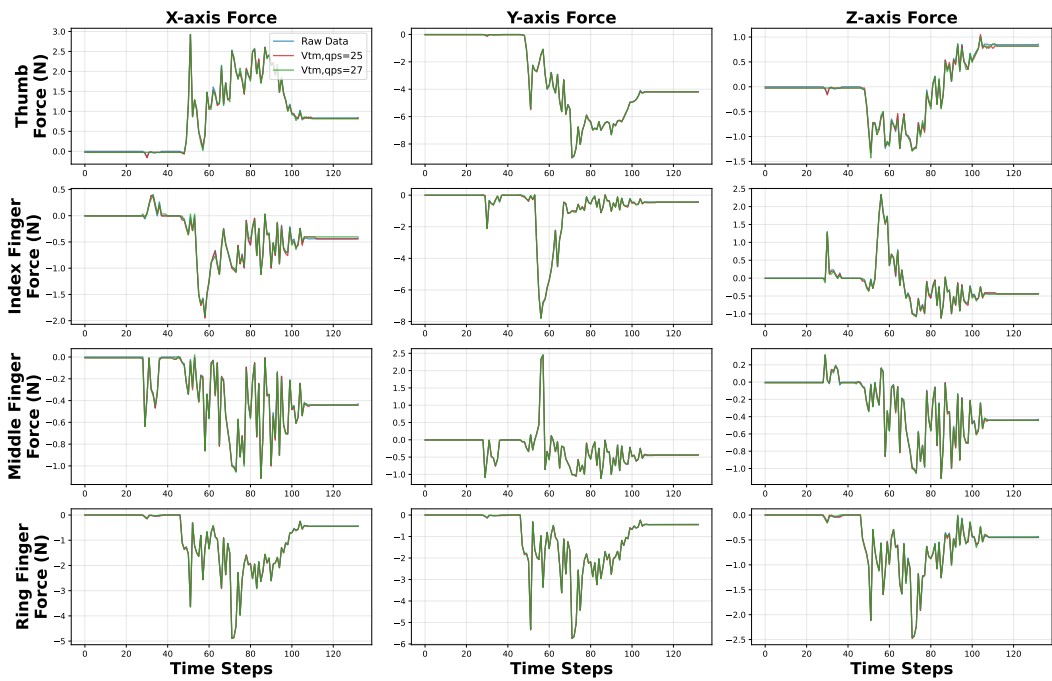

Figure 15: Visualization of tactile signals in the simulation experiments with VTM as the codec.

close the feedback loop for delicate tasks, enabling true physical understanding and interaction at a distance.

Third, to scale up in the field of embodied intelligence, the creation of large-scale training datasets necessitates efficient storage solutions. Specifically, Google introduced Open X-Embodiment Dataset, the largest open-source real robot dataset to date. It contains 1M+ real robot trajectories

(download size is **8965 GB**) O'Neill & Rehman (2024). While video compression is mature, specialized algorithms for tactile data remain underdeveloped, hindering our ability to build and manage the vast datasets required for training generalist robotic models. These pressing needs collectively motivate the establishment of a rigorous benchmark to advance the field of tactile data compression.

For the future work, we will develop a video-like tactile codec by retraining the tactile dataset using the latest neural video compression models, like DCVC-serier models.

## A.9 LLM USAGE STATEMENT

Large Language Models (LLMs) were not used during the research, experimentation, or analysis phases of this work. During the manuscript preparation, LLMs were used solely for minor grammar and language refinements. No content, ideas, or technical writing was generated by LLMs.