# OpenReview forum: "TaCo:  A Benchmark for Lossless and Lossy Codecs of Heterogeneous Tactile Data"
_ICLR.cc/2026/Conference — ICLR 2026 Poster_

### Official Review · Reviewer_RqzN · 2025-10-16

**Soundness:** 3
**Presentation:** 3
**Contribution:** 3
**Rating:** 8
**Confidence:** 4

**Summary:**

This paper introduces TaCo, the first large-scale benchmark for tactile data compression, evaluating 30 codecs (off-the-shelf and neural) on five heterogeneous tactile datasets. It also proposes two data-driven codecs, TaCo-LL (lossless) and TaCo-L (lossy), trained end-to-end on tactile data. The benchmark spans four task types: lossless storage, human visualization, classification, and dexterous grasping. Extensive experiments show that the proposed codecs outperform existing methods across all tasks.

**Strengths:**

Timeliness and Relevance
Tactile sensing is rapidly maturing, yet compression remains fragmented and under-studied. A unified benchmark is sorely needed;
TaCo fills this gap convincingly.
Comprehensive Evaluation
The paper tests 30 codecs on >250 k frames from five datasets covering vision-based (GelSight, DIGIT) and force-based sensors, providing the broadest coverage to date.
Novel Data-Driven Codecs
TaCo-LL and TaCo-L are, to my knowledge, the first codecs trained exclusively on tactile data. They achieve new state-of-the-art bit-rates (e.g., 0.447 bits/Byte on Touch-and-Go, 18× compression) while preserving task accuracy.

**Weaknesses:**

Limited Physical Interpretation of Distortion
PSNR and MS-SSIM are used for “human vision” evaluation, but tactile images are seldom viewed by humans; their perceptual space is task-dependent (e.g., shear vs. normal force). The paper does not validate that a 3 dB PSNR drop translates to an imperceptible change for either human fingers or downstream controllers. A small discrimination experiment (e.g., JND for contact edge localization) would strengthen the lossy compression claims.

Dataset Bias and Generalisation
All five datasets are collected on rigid, Lambertian objects. Soft or textured materials (foam, fabric, skin) exhibit elasto-plastic deformation and specularity that violate the stationary-signal assumption of both CNN and transformer codecs. The claim “superior on heterogeneous tactile data” is therefore premature. At minimum, the authors should report cross-dataset transfer (e.g., TaCo-L trained on Touch-and-Go, tested on ActiveCloth).
Baseline Fairness for Neural Codecs
Pre-trained image codecs (ELIC, LALIC, TCM) are evaluated zero-shot, whereas TaCo-L is fine-tuned on the target domain. This gives an unfair advantage. A fair comparison would fine-tune all neural baselines for the same number of epochs or report the “pre-trained → tactile” transfer gap.

Absence of Latency/Complexity Metrics
Real-time tactile feedback for tele-operation requires <1 ms end-to-end latency. The paper omits encode/decode runtime, GPU/CPU memory, and whether the arithmetic coder is single-pass. Without these numbers, a 190× compression ratio is meaningless for closed-loop control. Please add throughput (fps) vs. bitrate curves on an embedded GPU (Jetson Orin) or DSP.
Statistical Significance in Grasping Experiments
Table 6 reports success rates on 100 simulated objects. With only 8 deformable objects, the 95 % CI on the 62.2 % success rate is ≈ ±9 %, overlapping the baseline 63.8 %. The authors should use paired t-tests across seeds or bootstrap CIs to confirm that TaCo-L is not worse than uncompressed data.

**Questions:**

Figure 6 shows “visual” quality at 0.06–0.09 bpp, but tactile images are not human-interpretable. Replace with a tactile-specific error map (e.g., normal-force RMSE) or skip visual inspection altogether.
Table 3 uses BD-rate with HM-Intra as anchor. Please also quote absolute BPP at equivalent PSNR (e.g., 40 dB) so readers can judge real-world bandwidth (e.g., 0.04 bpp ≈ 1.2 Mbps at 30 fps).
Section 4.5 employs Isaac Sim tactile signals that are artificially sparse (hence 1000× ratio). Clarify that physical GelSight data achieve at most 22× (ObjTac) to avoid misleading roboticists.
Typos: “intra-frame compressors” header in Table 4 is misspelled “Jlrra-gt”; reference “Deletang et al. 2024” appears three times with inconsistent years.

---

> ### Author Response · Authors · 2025-11-22
> **Official Comment by Authors**
>
> We sincere thank the reviewer for the appreciation of our contribution and the careful review!
>
> **1. Limited Physical Interpretation of Distortion**
>
> Thanks for pointing it out. We have added more visualizations of raw and compressed tactile signals with four fingertips when grasping objects in the real-world and simulation with different lossy codecs to make the force loss more clearly and easier to understand the degradation causaed by lossy compression, as Fig.11-18 in the appendix.
>
> **2.Dataset Bias and Generalisation**
>
> Thanks for this comment. We have added the anlysis on cross-sensor and cross-object performanc evaluation in Appendix A.2 and A.3. Especially, we evaluate the performance of TaCo-LL and TaCo-L, trained using 70% of Touch and Go, and ObjTac, and then test the performance on a new dataset ActiveCloth. Our models still show the best performance among codecs, but the results also suggest that soft objects in Active Cloth are more difficult to compress than rigid objects. Among baseline neural codecs, we select the best and SOTA LALIC architecture on images, then adapt and retrain this model to show that learning tactile-specific priors is valuable.
>
> **3. Absence of Latency/Complexity Metrics**:
>
> Thanks for this comment. We have added detailed complexity comparison of TaCo-LL and TaCo-L with existing codecs, as Table 3 and Table 5, including Parameters, MACs, throughput KB/s and frame per second (FPS). Please kindly refer to the updated manuscript.
>
> **4. Other questions**
>
> (1) We will add the error map for a clear visual comparison in the updated manuscript. (2) We have added the explaination to table caption. For the reference, the bandwidth consumption of the anchor HEVC-intra is approximately 2Mbps at the quality of 40dB, which is calculated by $0.22$ bit per pixel $\times 640 \times 480 \times 30$fps$ \times 10^{-6}$ for Touch and Go dataset . (3) We have added the explaination in the Sec. 4.5 and checked the typos.
>
> Again, we thank the reviewer for the recognition on Timeliness and Relevance of our work, and please let us know if there is any additional question we can help address.

---

### Official Review · Reviewer_zy4w · 2025-10-23

**Soundness:** 2
**Presentation:** 3
**Contribution:** 1
**Rating:** 2
**Confidence:** 4

**Summary:**

This paper introduce TaCo, the first comprehensive benchmark for Tactile data Codecs.  It comprises five publicly tactile datasets, 30 codecs, and four tactile-related tasks. It  introduce two data-driven tactile codecs TaCo-LL and TaCo-L, and conducts experiment to validate their performance.

**Strengths:**

1. The proposed Toco benchmark is the first systematic benchmark specifically designed for tactile data compression.
2. The paper addresses the problem of tactile data compression, which is indeed an important yet overlooked issue.
3. The dataset encompasses a variety of tactile sensors.
4. The work evaluates the impact of tactile data compression on several downstream tasks.

**Weaknesses:**

1. The practical contribution of this work is rather limited. The proposed TaCo benchmark is essentially a simple aggregation of several existing datasets, without introducing any new data or labels. The proposed TaCo-L and TaCo-LL are in fact direct applications of two existing methods, DualComp-I and LALIC, trained on tactile data. Essentially, this work only integrates existing datasets to form a new dataset specifically for the tactile data compression task.
2. In both Figure 1 and the introduction, the paper highlights the significant intra-frame and inter-frame correlations in tactile videos, which should be a key distinction between tactile videos and natural RGB videos. However, disappointingly, the paper barely discusses this crucial aspect in its methodology, experiments, or metric design. Instead, it simply adopts existing approaches from natural image and video compression and applies them directly to the tactile domain.
3. An important distinction between tactile data and natural image or video modalities is that tactile sensing mainly serves object interaction. Therefore, validating performance only on material classification tasks or simple grasping tasks in simulation environments, which are not closely related to real tactile interaction, is insufficient to demonstrate its effectiveness. More real-world object manipulation experiments should be included.
4. A prominent issue in the tactile domain is the low diversity of available data and the significant heterogeneity among sensors. Does the proposed tactile data compression algorithm possess cross-sensor and unseen object generalization capabilities (the latter being particularly important)? Because in real-world applications, the objects that tactile sensors come into contact with can be highly diverse.
5. Another key point is that tactile data cannot exist independently of objects. Therefore, it is important to compare and discuss methods such as ObjectFolder, which use implicit neural representations to model objects. Compared with these methods, do tactile compression algorithms also offer advantages in terms of quality and efficiency?
6. Would tactile data compression negatively affect the spatially dependent information in tactile data? This aspect is crucial for tactile sensing, which is a highly fine-grained, sensitive, and localized modality during object manipulation.
7. When applying tactile compression algorithms to compress tactile data for downstream tasks, how does this differ from and what advantages does it have over using pretrained tactile encoders (such as UniTouch [1], T3 [2], or AnyTouch [3]) for feature preprocessing (which also can be seen as compressing)? What kinds of problems specifically require tactile compression algorithms rather than these pretrained encoder-based methods? Possible experimental results should be provided to support the claims, along with thorough and detailed discussions.
8. There are some confusing aspects in the experimental setup. For the Touch and Go dataset, since each recording segment contains many frames of the same object, the official split is based on collection trajectories. Why does this work not follow the official split and instead choose to perform a random split? In addition, the ObjectFolder dataset is a multimodal object dataset based on implicit neural representations, so the procedure for the material recognition task in this part needs to be explained in detail.

Overall, I believe the contribution of this work is quite limited. It simply combined several existing tactile datasets and revalidated existing techniques and experiences from image and video compression in the tactile domain, while overlooking many unique characteristics of tactile sensing. Therefore, considering the numerous issues present in the paper and the high quality standards of ICLR, I am inclined to reject this paper unless the authors can fully address these problems within a short period of time, including adding additional experiments and extensive discussions.



[1] Yang, Fengyu, et al. "Binding touch to everything: Learning unified multimodal tactile representations." *Proceedings of the IEEE/CVF Conference on Computer Vision and Pattern Recognition*. 2024.

[2] Zhao, Jialiang, et al. "Transferable tactile transformers for representation learning across diverse sensors and tasks." *arXiv preprint arXiv:2406.13640* (2024).

[3] Feng, Ruoxuan, et al. "Anytouch: Learning unified static-dynamic representation across multiple visuo-tactile sensors." *arXiv preprint arXiv:2502.12191* (2025).

**Questions:**

1. In the appendix's motivation section, it is mentioned that one key challenge is that tactile data from high-resolution sensor arrays on robotic hands can consume a large portion of the available bandwidth. I would like to know how this problem can be addressed using tactile compression algorithms, especially for vision-based tactile sensors that are essentially cameras.
2. There are already some LLMs capable of handling tactile inputs [1,2]. Why not consider using these methods instead of using LLaMA, which has never been exposed to tactile data?
3. The experimental results on YCB-Slide are extremely high, with the uncompressed baseline reaching 99%. Is there an issue with the data split in the experiment?



[1] Yang, Fengyu, et al. "Binding touch to everything: Learning unified multimodal tactile representations." *Proceedings of the IEEE/CVF Conference on Computer Vision and Pattern Recognition*. 2024.

[2] Yu, Samson, et al. "Octopi: Object property reasoning with large tactile-language models." *arXiv preprint arXiv:2405.02794* (2024).

---

> ### Author Response · Authors · 2025-11-22
>
> We sincere thank the reviewer for the appreciation of our contribution (eg. the first systematic benchmark of tactile data compression, address the tactile data compression problem, evaluation on several downstream tasks) and the detailed and careful review! Regarding to your concerns, we have added results in updated manuscript.
>
> **1. Practical Contribution**
>
> The dataset preparation is just a part of our work, and our primary contribution lies in constructing the first systematic benchmark for tactile data compression, to fill the gap to this field, which has already been recognized by your strengths part and other reviewers. TaCo-LL and TaCo-L are adapted from existing SoTA learning-based codecs, but retraining models can bring consistently superior performance over generic codecs across modalities, indicating that learning tactile-specific priors is nontrivial and valuable. It establishes a solid empirical foundation for future work on learning-based tactile compression and modality-aware compression.
>
> **2. Regarding intra- and inter- frame correlations**
>
> We would like to explain your concerns from two aspects.
>
> - `Similarity`: In fact, natural-scene RGB images also exhibit significant intra-frame (spatial) and inter-frame (temporal) correlation, which is precisely what gave rise to the key techniques of intra-prediction and inter-prediction in modern visual codecs (including MPEG, H.264, H.265, H.266 and learning-based visual codec). Therefore, **this is not a key distinction but a fundamental similarity**, and it is this commonality that motivates our use of the architecture of visual codecs for tactile data.
>
> - `Difference`: On the other hand, this intra- and inter- correlation only demonstrates that there is a large redundancy to be reduced for tactile data, while the extend of correlation or explicit data distributions between tactile data and RGB images are totally different, which drives us to retrain the model using the tactile dataset, which finally achieves the superior performance.
>
> **3. Regarding to object interaction**
>
> We have added the real-world object manipulation experiments in the appendix (section A.7 and A.8).
>
> As shown in Figure 12 ~ Figure 19, we can find that in fact, the compression algorithm itself does not affect the main properties and changing distributions of tactile data, so it will not have a catastrophic impact on the accuracy of the real-world manipulation task. One advantage of compression methods is mainly to save the space for storage, the bandwidth of transmission.
>
> **4. Cross-dataset and object generalization capabilities**
>
> We have added two new sections "A.2 cross-dataset compression performance" and  "A.3 cross-object compression performance" in the appendix.
>
> - When comparing Active Cloth and Touch and Go at the same resolution of $640\times480$, our TaCo-LL model, with 96M parameters, achieves the best performance on both datasets. It achieves 0.723 bit/Byte and 0.447 bit/Byte, corresponding to compression ratios of $11\times$ and $18\times$, respectively. The results also suggest that soft objects in Active Cloth are more difficult to compress than rigid objects, as deformable surfaces tend to generate more complex tactile data. When comparing ObjectFolder-1.0 and ObjectFolder-2.0 at the same resolution of $120\times160$, all the compression methods basically achieve consistent results and our TaCo-LL also achieve the best performance with 2.855 bit/Byte, corresponding to compression ratios of $2.8\times$.
>
> - These findings are further supported by the BD-Rate comparisons in Table 10, where TaCo-L consistently achieves the lowest BD-Rate across both ActiveCloth and ObjectFolder-2.0 datasets, outperforming state-of-the-art neural compressors such as ELIC, LALIC, and TCM.
>
> **5. Comparison with ObjectFolder**
>
> Thanks for this question. One great advantage of implicit neural representation (INR) is fast rendering, but the most critical issue is that INR is instance-adaptive, i.e. for each object, we need to train a new neural network to compress this object. The motivation of ObjectFolder is to construct a new dataset, where their object are deterministic, then they choose INR for pre-compression. But they are not a general-purpose compressors. Therefore, this paper did not include INR-based compression methods.
>
> **6. The impact of tactile data compression on spatially dependent information**
>
> Tactile data compression is inherently an optimization process for compact information representation, so it may negatively affect spatially dependent information. For the applications (such as sensitive teleoperation), which do not want to introduce this impact, lossless compression can be used.
>
> In this paper, we don't include the direct evaluation on spatial understanding or locallization tasks only using tactile data, since the baseline algorithm is not very clear. But this point can be further extended.

---

> ### Author Response · Authors · 2025-11-22
>
> **7. Difference between Tactile Codec and Pretrained  Tactile Encoders**
>
> -  `Technical Difference`: Tactile Codec involves the compression and decompression, for both reconstruction and understanding. Pretrained tactile encoders (including UniTouch, T3 and AnyTouch) are developed for modality-aligment, semantic understanding, which is a feature extractor, but do not consider for signal reconstruction.
>
> - `Application Difference`: Tactile data compression are essential when data acquisition and usage are decoupled in time or space, such as in teleoperation, edge-to-cloud transmission, or long-term storage. They ensure the tactile signal can be efficiently transmitted, stored, and later reused for diverse downstream tasks. Pretrained tactile encoders generate compact, task-specific features designed for semantic understanding or modality alignment. While effective for inference, these features are not intended for signal reconstruction and may not generalize across tasks that require access to the original tactile signal, such as force estimation or contact replay.
>
> **8. Experimental Setup**
>
> - `Train/test split`: For the Touch and Go dataset, while the official guideline recommends splitting by collection trajectories, it does not specify an exact train/test ratio or content. We followed this recommendation by grouping data at the trajectory level and applied a common 70%/30% split for training and testing. Each trajectory was then decomposed into individual frames, ensuring that all frames from the same trajectory are contained entirely within either the training or the testing set, avoiding any data leakage. We have added in the updated version.
>
> - `ObjectFolder usage`: For the material recognition task on the ObjectFolder-2.0 dataset, we follow the official rendering pipeline to generate tactile images from the implicit neural representations. ("python OF_render.py --modality touch ..."), detailed codes can refer to the official git \url{https://github.com/rhgao/ObjectFolder}.
>
> **Q1. The bandwidth consumption on robotic hands**
>
> High-resolution tactile sensors pose great bandwidth challenges. Tactile compression algorithms address this by drastically reducing data volume while preserving signal fidelity. For example, for a dense tactile array (e.g., 1140 taxels × 3 axes) sampled at 100 Hz, the raw data requires ~2.7 Mbps. After applying our lossless codec, the bandwidth is reduced to ~12.5 Kbps, equivalent to a 200× compression ratio with no information loss.
>
> For vision-based sensors like GelSight or DIGIT. They typically produce RGB images at frame rates of 30–100 Hz, with each frame being on the order of hundreds of kilobytes. For instance, a 640×480 RGB image at 8 bits per channel results in ~900 KB/s at 30 Hz, or ~3 MB/s at 100 Hz.
>
> **Q2. Usage of LlaMA**
>
> [Yang, CVPR'24] and [Octopi] are both for tactile data understanding, which is lossy and can not reconstruct the raw data, but we use LlaMA to generate the probability model, and use entropy coder for lossless compression.
>
>
> **Q3. Results on the YCB-Slide dataset**
>
> There is no issue with the train/test split on YCB-Slide (we strictly avoid overlap between training and testing samples). The high classification accuracy is due to the nature of the dataset: YCB-Slide contains only 10 object categories, each producing highly distinctive tactile patterns. Hence, the classification task on YCB-Slide is relatively easier than our other experiments.

---

### Official Review · Reviewer_ieBG · 2025-10-31

**Soundness:** 2
**Presentation:** 2
**Contribution:** 2
**Rating:** 6
**Confidence:** 4

**Summary:**

This paper introduces TaCo, a comprehensive benchmark for tactile data compression. TaCo evaluates 30 codecs (17 off-the-shelf, 13 neural) across five heterogeneous tactile datasets comprising over 250K samples.The benchmark measures both lossless and lossy compression, connecting performance to four downstream tasks: data storage, human visualization, material/object classification, and robotic grasping. Furthermore, the authors train two tactile-specific codecs, TaCo-LL (lossless) and TaCo-L (lossy), adapted from DualComp and LALIC, showing that modality-specific training can substantially improve compression efficiency and downstream task fidelity.

**Strengths:**

1. Addresses a clear and underexplored bottleneck.
TaCo fills a critical gap in the tactile perception community: the absence of a standardized, quantitative framework for evaluating tactile data compression.The scope—five tactile datasets, multiple codecs, and four downstream tasks—demonstrates strong engineering execution and community value.
2. Task-grounded evaluation beyond compression metrics.
The inclusion of real downstream metrics (classification, grasping) moves the discussion beyond raw rate–distortion numbers and demonstrates practical significance for embodied and robotic systems.
3. Evidence that tactile-specific retraining is beneficial.
The adapted models (TaCo-LL, TaCo-L) consistently outperform generic codecs across modalities, indicating that learning tactile-specific priors is nontrivial and valuable. This establishes a solid empirical foundation for future work on modality-aware compression.

**Weaknesses:**

1. Limited methodological originality.
 While the benchmark is extensive, the codec architectures themselves largely reuse existing designs (DualComp and LALIC) with dataset-specific retraining. The paper’s main novelty lies in benchmarking and empirical synthesis rather than proposing new modeling principles or theoretical insights into tactile compression.
2. Insufficient analysis of tactile data characteristics.
 The paper shows that tactile-specific codecs outperform generic ones but does not investigate the underlying statistical or structural differences that drive this improvement. Analyses such as entropy distribution, temporal redundancy, or spatial frequency spectra could provide insight into why certain codecs perform better on specific tactile modalities.
3. Lack of robustness and generalization analysis.
 The models are trained on a subset of datasets and evaluated on others, but there is no quantitative assessment of domain shift or variability (e.g., performance variance across seeds, datasets, or other sensor types such as vibrotactile, event-based, or thermal sensing are not included). Without such analysis, the extent to which the proposed codecs generalize beyond the training distribution remains unclear.
4. Missing runtime and deployment evaluation.
 Although model sizes are reported, the inference time such as KB/s or other metrics related to time efficiency is not analyzed. These aspects are important for understanding whether the proposed codecs are suitable for real-time robotic applications.

**Questions:**

1. What specific reasons motivated the choice of ObjectFolder (v1) instead of the newer ObjectFolder 2.0, which features higher scene diversity and richer material representations?
 Were there technical constraints (e.g., data volume, format consistency), or does ObjectFolder 2.0 differ in ways that make it unsuitable for this benchmark?
2. What specific modifications were made to DualComp and LALIC architectures when adapting them to tactile data?
 Please clarify in detail whether the entropy modeling, tokenization, or context mechanisms differ from the originals.
3. How were the grasping experiments conducted with respect to compression?
Section 4.5 describes a grasping simulation built in NVIDIA Isaac Sim using DexHand-13 with 11 tactile sensors, where pre-compressed tactile signals are fed into a tactile-aware RL controller. Could you provide additional details such as the tactile sampling rate, control frequency, and how compression latency was incorporated into the control loop?
 Moreover, robotic grasping tasks are particularly sensitive to real-world tactile performance—success rate, contact stability, and control frequency often differ substantially between simulation and physical setups. It would therefore be valuable to discuss or empirically examine how the proposed codecs might behave on real robotic hardware, even at small scale, to assess practical applicability beyond simulation.
4. Could you report the inference-time throughput (e.g., KB/s) and latency of TaCo-LL and TaCo-L compared with existing codecs?
 Since DualComp—one of the models adapted in this work—emphasizes runtime efficiency and reports detailed KB/s metrics, including similar results here would clarify whether the proposed tactile codecs preserve comparable efficiency. This metric is also particularly important for real-time robotic systems.
5. Have you examined how sensitive TaCo-LL and TaCo-L are to the training data distribution (e.g., varying tactile sensor types or object categories)?
 This could clarify whether performance gains arise from data diversity or from architectural suitability.

---

> ### Author Response · Authors · 2025-11-21
>
> We sincere thank the reviewer for the appreciation of our contribution (including the community value, task-grounded evaluation and evidence of retraining's benefits) and the careful review!
>
> Regarding to your questions, we have added new experiments to the updated manuscript.
>
> **1. Use of ObjectFolder-2.0**
>
> OF-1.0 and OF-2.0 are both collected using the same sensor Gelsight, then we just choose a small set and there is no technical constraints to extend to OF-2.0. We have already included in the updated version (A.2 section, Table 9, and Table 10 of the supplementary material).
>
> **2. Specific modifications to DualComp and LALIC**
>
> We add the detailed decription and figures to describe the modifications to DualComp and LALIC in the updated version and quote them here.
>
> - For TaCo-LL, the tokenization is conducted as shown in Figure 3. We divide the input into $16\times16\times3$ patches to preserve local spatial correlations. We then flatten the data in a raster-scan order. For visuo-tactile data, including Touch and Go, YCB-slide, ObjectFolder, SSVTP, the RGB values are sequentially expanded as sub-pixels $(R_1, G_1, B_1, R_2, G_2, B_2, \cdot\cdot\cdot)$. For three-axis force signals are processed as the color component and expanded as $(x_1, y_1, z_1, x_2, y_2, z_2, \cdot\cdot\cdot)$.
>
> - For TaCo-L, we follow the setup of LALIC and randomly crop or zero-pad the input tactile image to $256\times256$ resolution. Since the input tensor has three channels for both visuo-tactile data and force-tactile data, no tokenization is needed, as shown in the right part of Figure 3. The network architecture is adopted from the LALIC model~\citep{lalic} and the $g_a$ and $g_s$ consist of four downsampling and upsampling operations, respectively.
>
> **3. Grasping with Compression**
>
> First, we have added the explaination on how to compress tactile signals in a tactile-aware RL controller in the update version and quote them below. Second, we have also added the real-world experiments, and included them in section A.7 of the supplementary material.
>
> > The simulation environment for the reinforcement learning controller operates at a control frequency of 100 Hz, which is determined by the simulation time step of 0.01 seconds.
> > (1)The tactile sensors are updated at every simulation step, resulting in a tactile sampling rate of 100 Hz.
> > (2)The overall latency of the control loop is approximately 0.01 seconds, plus the time required for policy inference. The policy inference is performed using an ONNX model, and the inference time is logged during execution. If the inference time exceeds the simulation time step, the control frequency may decrease, and the latency would increase accordingly.
> > (3) When the combined codec and inference latency approximately equals the simulation update interval, the additional delay introduced to the simulation environment becomes negligible, as it aligns with the natural timing cycle of the control loop.
> > However, in the current implementation, the control command is applied in the same simulation step after inference, so the latency is primarily determined by the simulation step and the inference time.
>
>
> **4. Complexity Performance**
>
> Thanks for this comment. We have added detailed complexity comparison of TaCo-LL and TaCo-L with existing codecs, as Table 3 and Table 5, including Parameters, MACs, throughput KB/s and frame per second (FPS). Please kindly refer to the updated manuscript.
>
> **5. Varying tactile sensor types or object categories to compression performance**
>
> We have added the anlysis on cross-sensor and cross-object performanc evaluation in Appendix A.2 and A.3. Please kindly refer to the updated manuscript. We have also included the analysis of tactile data characteristics in A.4, and quote them here.
>
>
> Again, we thank the reviewer for the recognition of our work, and please let us know if there is any additional question we can help address.

---

### Official Review · Reviewer_bciL · 2025-10-31

**Soundness:** 2
**Presentation:** 2
**Contribution:** 3
**Rating:** 4
**Confidence:** 3

**Summary:**

This paper proposed TaCo as the first comprehensive benchmark for evaluating tactile data codecs.
TaCo evaluates existing classical and neural codecs on lossless storage, human visualization, material and object classification, and dexterous robotic grasping.
The authors also proposed TaCo-LL (lossless) and TaCo-L (lossy) models and achieved superior performance on the benchmark.

**Strengths:**

The motivation is important for the community. Bandwidth requirement has been one of the main bottlenecks for vision-based tactile sensors nowadays, whose temporal resolution is usually limited despite the high spatial resolution. A benchmark for tactile codecs is necessary to push forward the research in this direction. The benchmark design is comprehensive and covers a wide range of tasks that are important for tactile data usage, making it a valuable resource for the community.

**Weaknesses:**

1. Still not enough discussion/experiment to support the motivation. To decrease the bandwidth requirement for tactile hands in reality, a discussion/metric on whether/how these methods are computationally feasible on the sensor side hardware is necessary. Without that, the motivation of this direction is vague.
2. Lack details about the method itself. "Symbol" predicted by $f_a$ is not defined anywhere nor is its dimension or quantization level for AE clarified. Neural nets used in either setting not explained. Lack high level explanation in Sec 3.2.2.
3. Writing issues, including but not limited to: Typos; Citation formats messed up; Missing full stops; Lack of explanation of abbreviations; Eq. 1 is incomplete; Table 2 lacks explanation on what does your -12M/48M/96M mean, and why some of the results has a ratio after it but some doesn't. How is Table 4 related to the topic of the paper is unclear.
4. Lack qualitative results. As the only figure in the paper that shows qualitative results, Fig 6 is way too subtle that I cannot see the difference. Also I highly doubt the quality of tactile signals you generated within Isaac Sim. On the codec aspect, the Sim2Real gap for tactile simulation is not ignorable in this case. The lack of artifacts and much simpler optical setup could make signal behavior far from real world scenarios. On simulation itself, can you show qualitative examples to prove the results are reasonable, especially for those deformable-deformable contacts you claimed?
5. For the quantitative results, especially the dexterous hand grasping experiment, what's the purpose of introducing $s_{disturb}$ experiment? Can you explicitly explain that in your experiment design section? I also don't think the result is "comparable" to VTM-Intra.

**Questions:**

Covered in "Weaknesses".

---

> ### Author Response · Authors · 2025-11-21
>
> We sincere thank the reviewer for the appreciation of our contribution to the community value, and the careful review! Regarding to your questions, we have added new experiments to the updated manuscript.
>
> **1. Computation Complexity Performance**
>
> Thanks for this comment. We have added detailed complexity comparison of TaCo-LL and TaCo-L with existing codecs, as Table 3 and Table 5, including Parameters, MACs, throughput KB/s and frame per second (FPS). Please kindly refer to the updated manuscript.
>
> **2. Detailed Decription about our method**
>
> We add the detailed decription and figures to describe the modifications to DualComp and LALIC in the Sec. 3.2.2 of updated version and quote them here.
>
> - For TaCo-LL, the tokenization is conducted as shown in Figure 3. We divide the input into $16\times16\times3$ patches to preserve local spatial correlations. We then flatten the data in a raster-scan order. For visuo-tactile data, including Touch and Go, YCB-slide, ObjectFolder, SSVTP, the RGB values are sequentially expanded as sub-pixels $(R_1, G_1, B_1, R_2, G_2, B_2, \cdot\cdot\cdot)$. For three-axis force signals are processed as the color component and expanded as $(x_1, y_1, z_1, x_2, y_2, z_2, \cdot\cdot\cdot)$.
>
> - For TaCo-L, we follow the setup of LALIC and randomly crop or zero-pad the input tactile image to $256\times256$ resolution. Since the input tensor has three channels for both visuo-tactile data and force-tactile data, no tokenization is needed, as shown in the right part of Figure 3. The network architecture is adopted from the LALIC model and the $g_a$ and $g_s$ consist of four downsampling and upsampling operations, respectively.
>
> **3. Writing issues**
>
> Thanks for pointing it out. We have corrected Eq. 1, and add the explanation to Table2, where 12M/48M/96M denotes the model parameter and only for the best and second best performance we further list the compression ratios relative to the uncompressed data to make the results more clearly. We have also moved the Table 4 to the appendix.
>
> **4. Lack qualitative results**
>
> In section A.7 of the supplementary material, we provide a set of qualitative comparisons between real-world and simulated grasping experiments. Specifically, we analyze the force–tactile variation curves across four fingertips, demonstrating consistent signal behavior between simulated and real settings.
>
> Moreover, we evaluate how four commonly used compressors affect the tactile signal reconstruction and its impact on downstream physical interaction, particularly the force profiles of major hand joints. These results not only validate the realism of our simulated tactile signals but also highlight the importance of appropriate compression in preserving physically meaningful tactile information.
>
> **5. The purpose of introducing the dexterous hand grasping experiment**
>
> We have revised the experimental design section in the main paper to clarify the motivation behind the dexterous hand grasping experiment. Specifically, many contact-rich manipulation algorithms for dexterous hands rely heavily on high-fidelity tactile signals. In real-world deployment scenarios, however, tactile data compression may affect the downstream performance of such algorithms. Therefore, we introduce this experiment to evaluate the impact of tactile compression quality on a realistic, task-driven benchmark.
>
> Additionally, we have rephrased our comparison with VTM-Intra to avoid any misleading claims. While our method is within 1% of VTM's performance in terms of task success rate, it achieves this with significantly higher compression efficiency.

---

### Author Response · Authors · 2025-11-25
**Kind Follow-up on Review Status and Remaining Concerns – Paper 7210**

Dear AC and reviewers,

We thank the AC and reviewers for their time and insightful feedback. First of all, we would like to summarize the core contributions of our work.
- This paper introduces TaCo, **the first systematic benchmark for tactile codec**, which is an very important issue for robotics and potential immersive applications but now is largely overlooked. It comprises five diverse tactile datasets, 30 evaluated codecs, and four downstream tasks, thereby **addressing a significant gap** in the field of tactile data compression.
- Building upon this foundation, we propose two data-driven codecs: **TaCo-LL** for lossless and **TaCo-L** for lossy compression, both of which achieve **state-of-the-art performance**. This success underscores the value of learning tactile-specific priors for the broader multi-modal data compression community.

We are encouraged that all four reviewers basically recognized these key contributions.

In response to the reviewers' concerns, we have substantially revised the manuscript in the following main aspects:

-  **Complexity/Runtime Evaluation** (Raised by Reviewers ‘’bciL“, ''ieBG'', ''RqzN''):

    We have added detailed evaluations of compression speed (FPS) and bandwidth usage (KB/s) in Table 3 (for TaCo-L/LL) and Table 5 (for other codecs).

- **Generalization to Unseen Data** (Raised by Reviewers "zy4w", ''ieBG'', ''RqzN''):

    We have added cross-object and cross-dataset experiments in Appendix A.2 and A.3, testing on two new datasets (ActiveCloth and ObjectFolder 2.0) and visualizing the compression behavior across (soft/rigid, seen/unseen) objects.

- **Simulated/Real-world Grasping Results and Physical Interpretation of Distortion** (Raised by Reviewers "zy4w", ''ieBG'', "bciL", ''RqzN''):

    We have analyzed how compression affects downstream grasping in both simulation and real-world settings (Appendix A.4, Fig.11–Fig.19), by adding more visualizations to illustrate the physical interpretation and we have re-evaluated the human-perception distortion (Fig. 8) and the charateristic of tactile data (Appendix A.5 and Fig. 5).

- **Detailed Explanations** (Raised by Reviewers "zy4w", ''ieBG'', "bciL", "RqzN"):

    We have revised and expanded the descriptions of the methods and experimental setup, including task definitions, tactile data tokenization strategies, and experiment configurations throughout the main text, as seen in the newly highlighted (red) revisions. Furthermore, we have clarified the difference between the tactile codec and pretrained tactile encoders, which are targeted for different research fields and scopes,  and the bandwidth saving on robotic hands. We have answered all the concerns from all the reviewers point-by-point.

Thank ACs again for your great efforts to review this paper. We hope our revision has addressed all concerns and this work could lay a foundation for further tactile data compression and multimodal applications.

Best regards,

Author Team of Submission 7210

---

### Meta-Review · Area_Chair_T2gf · 2025-12-28

**Summary:**

This submission introduces TaCo, a systematic and much-needed benchmark for tactile data compression. While tactile sensing is foundational for embodied AI, the community has lacked a standardized way to evaluate how compression affects downstream tasks like grasping or material classification. Reviewer bciL primarily concerned with the lack of hardware-side feasibility metrics and the physical realism of the simulated tactile signals. Reviewer ieBG questioned the methodological novelty of the proposed codecs and the depth of analysis regarding tactile-specific data characteristics. Reviewer zy4w expressed strong reservations regarding the practical contribution, arguing the work was largely an aggregation of existing datasets without sufficient real-world manipulation experiments. Reviewer RqzN highlighted the need for more granular evaluation of computational complexity and runtime performance. I have decided to accept this work because it establishes a foundational framework for an essential problem. the authors’ rebuttal successfully introduced new real-world data, cross-dataset generalization tests, and hardware-relevant metrics that significantly strengthen the benchmark’s utility.

**Reviewer Concerns:**

The authors provided a detailed rebuttal that addressed several concerns, and some issues remain unsolved:

Addressed:
+ Complexity and Runtime: The authors added detailed evaluations of compression speed (FPS) and bandwidth usage ($KB/s$), directly addressing concerns about real-time robotic applicability.
+ Generalization: New cross-object and cross-dataset experiments were conducted using ActiveCloth and ObjectFolder 2.0, proving that the learned codecs can generalize beyond their training distributions.
+ Physical Realism: The authors provided qualitative force-tactile variation curves from real-world grasping, bridging the Sim2Real gap by showing consistent signal behavior between simulated and physical environments.
+ Methodological Details: Revisions clarified the tokenization strategies and the specific architectural adaptations of the TaCo-LL and TaCo-L models

Outstanding:
- Architectural Novelty: As noted by Reviewers ieBG and zy4w, the core codec architectures are largely adapted from existing image/video models (DualComp and LALIC) rather than being entirely new modeling paradigms.
- Implicit Neural Representations (INR): While the authors explained why INR-based methods like ObjectFolder are not general-purpose codecs, a direct performance comparison remains absent from the current benchmark.
- Spatial Dependency: The specific impact of lossy compression on fine-grained spatial localization tasks remains a point for future extension.

**Reviewer Scores:**

Based on the above issues, I think Reviewer bciL will increase to 6, the other three will keep their scores, finally 2,6,6,8.

---

### Decision · Program_Chairs · 2026-01-26

Accept (Poster)